# El Niño-like teleconnection increases California precipitation in response to warming

Robert J. Allen[1] & Rainer Luptowitz[1]

Future California (CA) precipitation projections, including those from the most recent Climate Model Intercomparison Project (CMIP5), remain uncertain. This uncertainty is related to several factors, including relatively large internal climate variability, model shortcomings, and because CA lies within a transition zone, where mid-latitude regions are expected to become wetter and subtropical regions drier. Here, we use a multitude of models to show CA may receive more precipitation in the future under a business-as-usual scenario. The boreal winter season-when most of the CA precipitation increase occurs-is associated with robust changes in the mean circulation reminiscent of an El Niño teleconnection. Using idealized simulations with two different models, we further show that warming of tropical Pacific sea surface temperatures accounts for these changes. Models that better simulate the observed El Niño-CA precipitation teleconnection yield larger, and more consistent increases in CA precipitation through the twenty-first century.

[1] Department of Earth Sciences, University of California Riverside, Riverside, California 92521, USA. Correspondence and requests for materials should be addressed to R.J.A. (email: rjallen@ucr.edu).

In response to increasing concentrations of greenhouse gases (GHGs), climate models from the Coupled Model Intercomparison Project (CMIP) versions 3 and 5 indicate decreases in precipitation in the subtropics and increases in middle to high latitudes[1,2]. California lies near this transition zone, which contributes to the relatively large uncertainty in future projections of CA precipitation[3,4]. Significant differences between CMIP3 and CMIP5 twenty-first century precipitation projections in central and southern California exist, with CMIP5 models tending to yield a more consistent increase[5]. This was related to an eastward extension of the upper level winds in the east Pacific, which was suggested to shift the storm track towards the California coast, promoting an increase in precipitation. Although Niño 3.4 sea surface temperatures (SSTs) were not found to exhibit a strong correlation with CA precipitation projections, only a portion of the CMIP5 ensemble was available. A subsequent study[6] looked at the CMIP5 model ensemble uncertainty in this region, and also explicitly sought an explanation in tropical Pacific SST. It found that the leading mode affecting the inter-model range of CA precipitation projections under global warming was related to a more complex large-scale pattern of SST changes with little expression in the equatorial cold tongue region. A second mode of North Pacific inter-model precipitation change differences was related to SST patterns that included a substantial El Niño-like component, but this had relatively modest expression in California rainfall.

Although future projections yield a zonal mean poleward (and upward) shift of the storm tracks[7], as well as storm track weakening in the Pacific[8,9], there is some evidence that storm track activity in the east Pacific may increase[10]. Part of this inter-model spread is related to model differences, and part is related to internal climate variability[11]. Furthermore, the cause of these dynamical responses remains unknown.

Here we use a multitude of models, including CMIP5 (ref. 12) and the Community Earth System Model (CESM) Large Ensemble (LENS) Project[13] to show CA may receive more precipitation in the future under a business-as-usual scenario. During Northern Hemisphere winter, when most of the CA precipitation increase occurs, models simulate robust changes in the mean circulation reminiscent of an El Niño teleconnection. This includes weakening of the Walker circulation, a poleward propagating Rossby wave that originates in the tropical central/eastern Pacific, a southeastward shift of the upper level winds and an increase in storm track activity in the east Pacific, and an increase in CA moisture convergence. Idealized simulations with two different models: the atmosphere component of CESM, the Community Atmosphere Model version 5 (CAM5)[14] and the Geophysical Fluid Dynamics Laboratory Atmospheric Model version 3 (GFDL AM3)[15], show that the increase in CA precipitation is associated with tropical Pacific SST warming. CMIP5 models that better simulate the observed El Niño-CA precipitation teleconnection yield larger, and more consistent increases in CA precipitation through the 21st century.

## Results

**Community earth system model large ensemble.** Figure 1 shows a spatial map of the annual (ANN) and December–January–February (DJF) CESM LENS (Methods) ensemble mean twenty-first century precipitation trend. On the basis of the annual mean, CA shows an increase of 0.22 mm day$^{-1}$ century$^{-1}$, significant at the 99% confidence level, with a range from ~0 to 0.48 mm day$^{-1}$ century$^{-1}$. CA precipitation increases more from south to north, with precipitation increasing by 0.03 mm day$^{-1}$ century$^{-1}$ in southern California (32.0–34.9°N; 239.4–245.6°E), 0.22 mm day$^{-1}$ century$^{-1}$ in central California (34.9–38.6°N; 236.9–243.1°E), and 0.41 mm day$^{-1}$ century$^{-1}$ in northern

California (38.8–42.4°N; 235.6–240.6°E). The latter two trends are significant at the 99% confidence level.

Most of the California precipitation increase occurs during DJF, where the entire region shows an increase of 0.94 mm day$^{-1}$ century$^{-1}$, with a range from ~0 to 1.9 mm day$^{-1}$ century$^{-1}$. As with the annual mean, precipitation increases from south to north-by 0.34 in southern California, 1.0 in central California, and 1.51 mm day$^{-1}$ century$^{-1}$ in northern California, all significant at the 99% confidence level. Also included in Fig. 1 is the percentage of realizations that yield an increase or decrease in precipitation. Grid points for which 25 out of 40 realizations (~65%) agree on sign pass a binomial test to reject the null hypothesis of equal probability of positive or negative sign at the 95% confidence level. During DJF, all but a handful of CA grid boxes (those in the southeastern corner) pass this test, indicating a high degree of consistency across realizations.

**Coupled model intercomparison project version 5.** Based on 38 CMIP5 Representative Concentration Pathway (RCP) 8.5 models (78 realizations in total; Supplementary Table 1), the CA annual mean precipitation trend ranges from −0.68 to 0.61, with a multi-model mean of 0.06 mm day$^{-1}$ century$^{-1}$ (significant at the 95% confidence level). Fifty-five per cent of the trends yield an increase in CA ANN precipitation. During DJF, the CMIP5 CA mean precipitation trend ranges from −1.14 to 2.62, with a multi-model mean of 0.51 mm day$^{-1}$ century$^{-1}$ (significant at the 99% confidence level). 71% of the trends yield an increase in CA DJF precipitation. These results show that CMIP5 models, relative to CESM LENS, yield a larger range (by at least a factor of two) of future CA precipitation projections, including significant drying. Thus, although internal climate variability accounts for some of the CMIP5 (and CESM LENS) spread, model differences appear to account for most of the spread.

It is well known that El Niño/La Niña accounts for some of the interannual variability in CA precipitation. Observations from 1948/49 to 2014/15, including precipitation from National Atmospheric and Oceanic Administration's Precipitation Reconstruction, show the correlation between Niño 3.4 SSTs (5S–5N; 190-240E) and CA DJF precipitation is 0.36, which is significant at the 99% confidence level. Similar values are found using alternative precipitation data sets, including the Global Precipitation Climatology Project, where the corresponding correlation from 1979/80 to 2014/15 is 0.40. During winter, in response to warmer central/eastern tropical Pacific SSTs, the El Niño teleconnection is associated with weakening of the Walker circulation and an eastward shift of deep convection and associated diabatic heating, which excites a Rossby wave response that alters the extratropical circulation[16–19]. This includes a southeastward shift of the Pacific jet stream, which guides more extratropical cyclones toward California[20–22]. All but one (FGOALS-g2) CMIP5 model-realization and all CESM LENS realizations shows warming of Niño 3.4 SSTs through the twenty-first century. Furthermore, 70% of the CMIP5 model-realizations and all CESM LENS realizations show enhanced warming of the tropical eastern Pacific (5S–5N; 190-240E) relative to the western Pacific (5S–5N; 120-170E). These results suggest a possible shift of the tropical Pacific to a more El Niño-like background state[23]. Many CMIP5 models, however, are unable to reproduce the observed correlation between Niño 3.4 SST and CA DJF precipitation. On the basis of the CMIP5 RCP8.5 models, this correlation (based on detrended time series) ranges from −0.12 in MIROC-ESM to 0.58 in GFDL-CM3 (Supplementary Table 1).

Sub-selecting the CMIP5 models that yield a detrended CA DJF precipitation versus Niño 3.4 SST correlation of at least

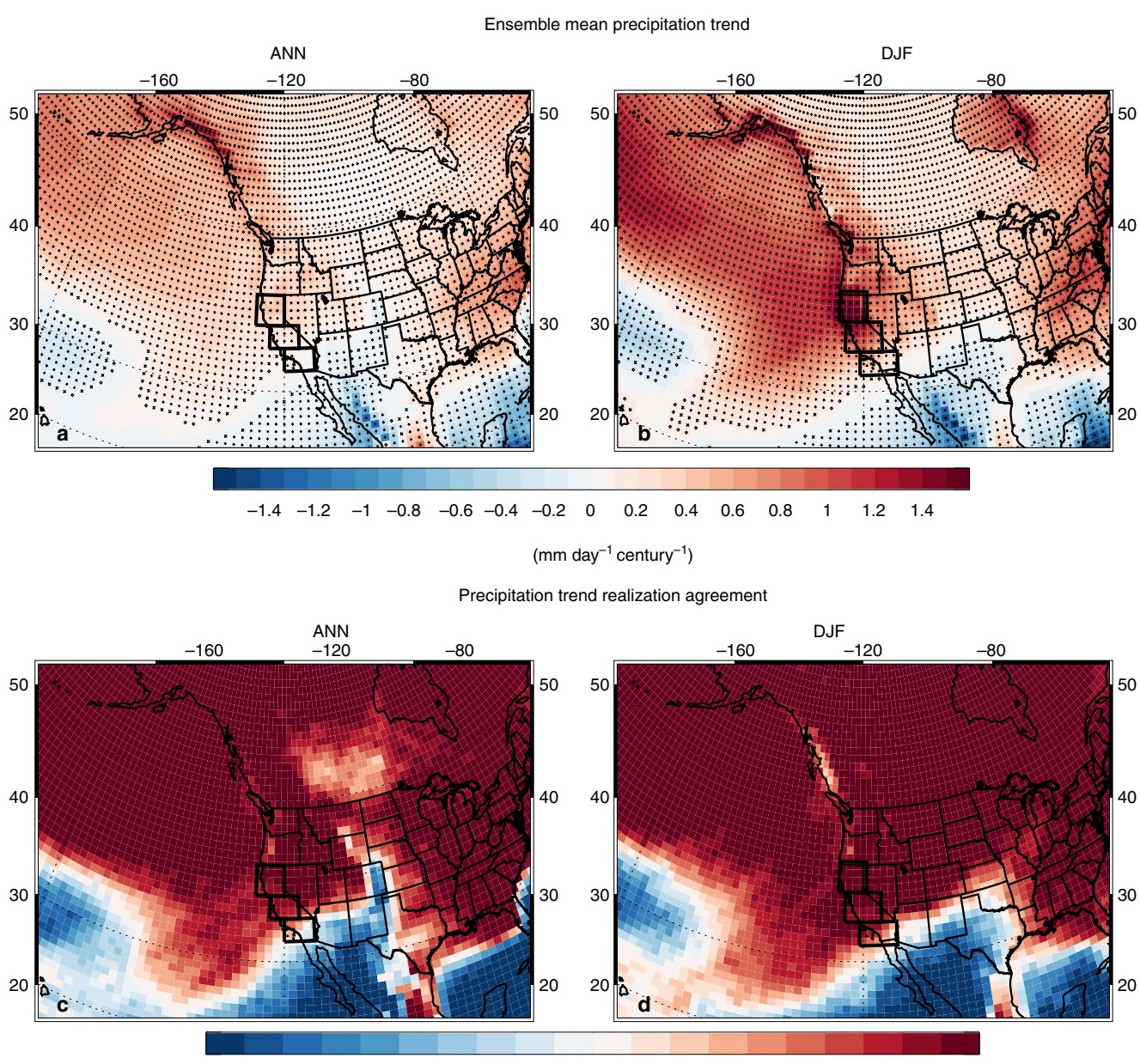

**Figure 1 | CESM LENS 2006–2100 precipitation change.** Ensemble mean (**a**) annual (ANN) and (**b**) DJF mean precipitation trend (mm day$^{-1}$ century$^{-1}$); (**c**) ANN and (**d**) DJF mean precipitation trend realization agreement (%). Symbols in **a**,**b** represent trend significance at the 90% (diamond), 95% (X) or 99% (+) confidence level, accounting for autocorrelation. Warm (cold) colours in **a**,**b** represent an increase (decrease) in precipitation. Warm (cold) colours in **c**,**d** show the per cent of realizations that yield an increase (decrease) in precipitation. Also included are the three regions comprising California, denoted with thick black lines.

0.30 yields 14 models (29 realizations in total), including CESM1-CAM5 (the CMIP5 version of CESM LENS). Using this CMIP5 subset, which we refer to as 'CMIP5 HIGH-r', we find larger, and more consistent, increases in CA precipitation (Fig. 2 and Supplementary Fig. 1). For example, the ensemble mean DJF (ANN) precipitation increase in this CMIP5 subset is 0.84 (0.16) mm day$^{-1}$ century$^{-1}$, both significant at the 99% confidence level. Furthermore, 79% (72%) of the model-realizations yield an increase in DJF (ANN) CA precipitation. CMIP5 HIGH-r also better reproduces the dynamical teleconnection associated with Niño 3.4 SST warming, including weakening of the Walker circulation, the increase in divergence, Rossby wave generation

(Methods) and Rossby wave response in the tropical central/eastern Pacific, the southeastward shift of the upper levels winds and the increase in storm track activity in the east Pacific (Supplementary Figs 2 and 3).

In contrast, sub-selecting the CMIP5 models that yield a CA DJF precipitation versus Niño 3.4 SST correlation <0.20 (12 models and 17 realizations), which we refer to as 'CMIP5 LOW-r', yields an ensemble mean DJF (ANN) precipitation trend of 0.09 (−0.15) mm day$^{-1}$ century$^{-1}$, the latter (negative) trend significant at the 99% confidence level. Furthermore, only 59% (24%) of the model-realizations yield an increase in DJF (ANN) CA precipitation. A $t$-test for the difference of means shows that these two model

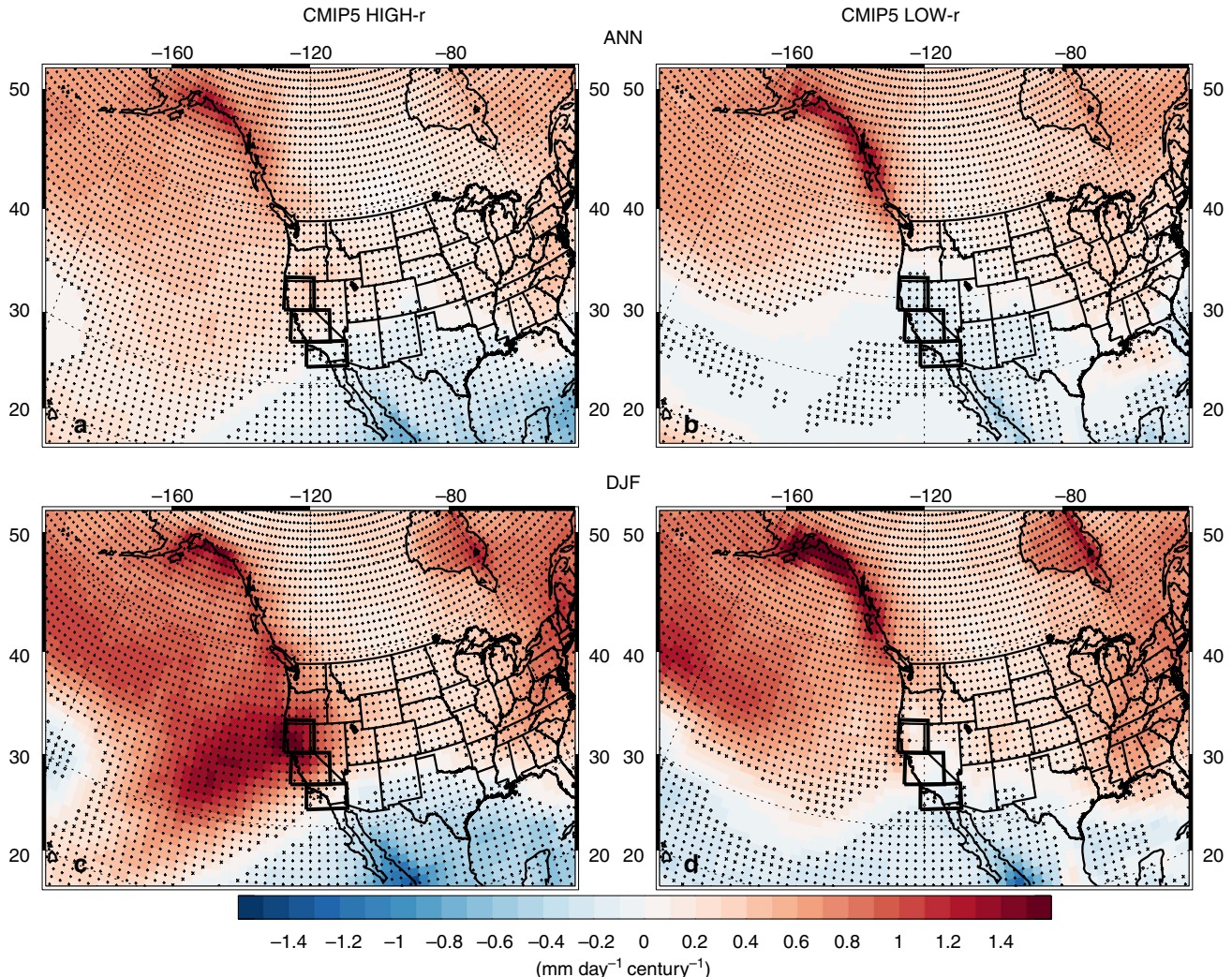

**Figure 2 | CMIP version 5 2006–2100 precipitation change.** Ensemble mean (**a,b**) annual (ANN) and (**c,d**) DJF precipitation trend for two CMIP5 model subsets. (left) The model subset that yield a detrended DJF Niño 3.4 SST versus California precipitation correlation of at least 0.30 (CMIP5 HIGH-r); (right) the model subset that yield a corresponding correlation less than 0.20 (CMIP5 LOW-r). Trend units are mm day$^{-1}$ century$^{-1}$. Symbols represent trend significance at the 90% (diamond), 95% (X) or 99% (+) confidence level, accounting for autocorrelation.

subsets are significantly different at the 99% confidence level based on both DJF and ANN CA precipitation trends. Similar results are obtained with different RCPs (Supplementary Fig. 4) and 1% $CO_2$ experiments (Supplementary Fig. 5), and if we slightly modify the correlation threshold used to define CMIP5 HIGH-r and LOW-r models (Supplementary Information). Thus, CMIP5 models that better reproduce the observed Niño 3.4 SST and CA precipitation teleconnection tend to yield larger, and more consistent, increases in CA precipitation through the twenty-first century.

We note that high correlation between CA precipitation and Niño 3.4 SSTs alone does not guarantee a more realistic response. Overly high correlation would also be unrealistic, and high correlation can go with unrealistic amplitude response. Some models exhibit larger than observed regressions of CA precipitation onto Niño 3.4 SSTs[24]. Furthermore, some models have a storm track that hits the US West Coast too far south. Supplementary Table 2 shows late twentieth century DJF correlations, regression coefficients, and climatological CA precipitation by region for CMIP5 models, along with the observed values and their 1-sigma uncertainty. The CA precipitation sensitivity to Niño 3.4 SSTs, and regional climatologies, for many models falls outside the observed range (Supplementary Information). We define an alternative

'HIGH-r' model subset that satisfies the following criteria: (1) late twentieth and twenty-first century correlations between DJF CA precipitation and Niño 3.4 SSTs are significant at the 90% confidence level; (2) late 20th century DJF CA precipitation versus Niño 3.4 SST regression coefficient falls within 1-sigma of the observed range; and (3) late 20th century DJF CA precipitation climatologies fall within 1-sigma of the observed range. These criteria result in three models, with 12 realizations. The ensemble mean twenty-first century DJF (ANN) CA precipitation trend in these models is 0.91 (0.24) mm day$^{-1}$ century$^{-1}$, both significant at the 99% confidence level. 83% (83%) of the realizations yield an increase in DJF (ANN) CA precipitation. These values are very similar to those based on our original correlation-based threshold. Moreover, the tropical and extratropical dynamical responses in these three models (not shown) are similar to those that will be discussed. Thus, we continue to find significant and robust increases in twenty-first century CA precipitation, particularly in models that better simulate CA precipitation statistics.

Figure 3 further supports the importance of Niño 3.4 SST warming to future increases in CA precipitation. On the basis of observed DJF seasonal variations, the spatial correlation between CA precipitation and SSTs yields a significant El Niño-like pattern,

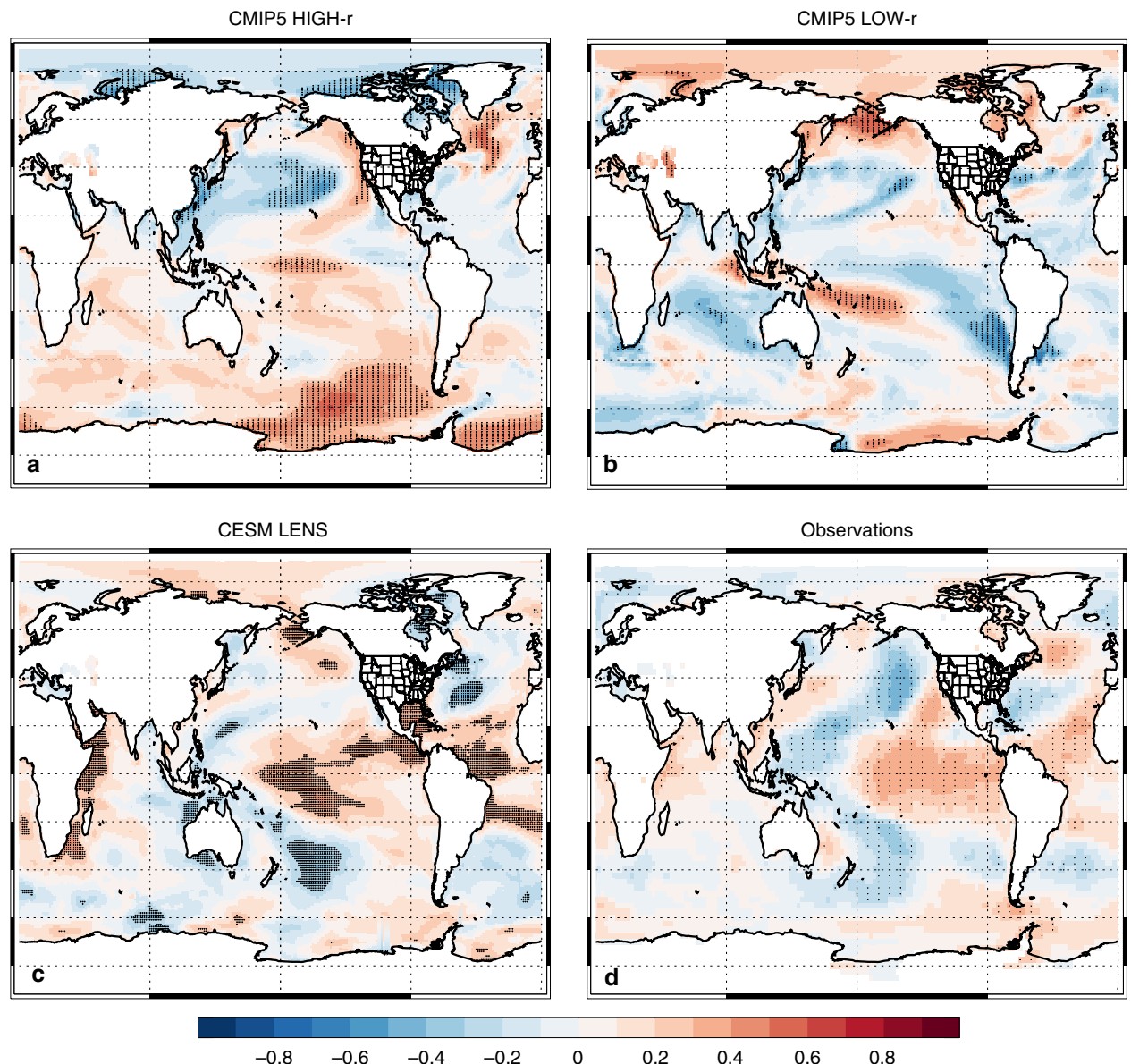

**Figure 3 | DJF California precipitation versus SST trend correlation map.** 21st century trend correlation between California (CA) precipitation and SST across individual realizations for the CMIP version 5 (CMIP5) model subset that yields a detrended DJF Niño 3.4 SST versus CA precipitation correlation of (**a**) at least 0.30 (CMIP5 HIGH-r), (**b**) <0.20 (CMIP5 LOW-r) and (**c**) the CESM LENS. (**d**) The observed 1948/49 to 2014/15 DJF correlation based on interannual variations, including precipitation from NOAA Precipitation Reconstruction and SSTs from the NCEP/NCAR Reanalysis. Symbols represent significance at the 90% (diamond), 95% (X) or 99% (+) confidence level, accounting for autocorrelation. NCEP/NCAR, National Center for Environmental Prediction/National Center for Atmospheric Research; NOAA, National Atmospheric and Oceanic Administration.

with more CA precipitation associated with warmer Niño 3.4 SSTs. Similarly, both CMIP5 HIGH-r and CESM LENS yield a similar spatial correlation, but based on twenty-first century DJF trends in CA precipitation and SSTs. In particular, both show significant trend correlations near the Niño 3.4 SST region. Thus, realizations that yield a larger increase in DJF CA precipitation are also associated with larger warming of the tropical central/eastern Pacific SSTs (that is, an El Niño like pattern). We note that CMIP5 LOW-r does not reproduce this relationship. Similar results are obtained for 1% $CO_2$ experiments (Supplementary Fig. 6).

**Dynamical mechanisms.** We now focus on the wintertime (DJF) change in CA precipitation, and investigate possible dynamical mechanisms associated with the increase. At monthly timescales

(>10 days), the atmospheric moisture budget indicates that precipitation, evaporation and net moisture flux into/out of an atmospheric column through its lateral boundary are in a balance[25]. The CESM LENS ensemble mean increase in DJF CA precipitation is divided into a DJF moisture flux convergence (MFC) increase of 0.69 mm day$^{-1}$ century$^{-1}$, and an increase in evaporation of 0.23 mm day$^{-1}$ century$^{-1}$ (Supplementary Table 3). All of the MFC increase is related to an increase by the mean flow, as the MFC mean trend is 0.72 and the MFC transient trend is $-0.04$ mm day$^{-1}$ century$^{-1}$. Further decomposing the MFC mean into its dynamic and thermodynamic components (Methods) shows that all of the increase in the MFC mean is related to dynamical processes[26]. The dynamic component increases by 0.80, whereas the thermodynamic component decreases by $-0.10$ mm day$^{-1}$ century$^{-1}$. Thus,

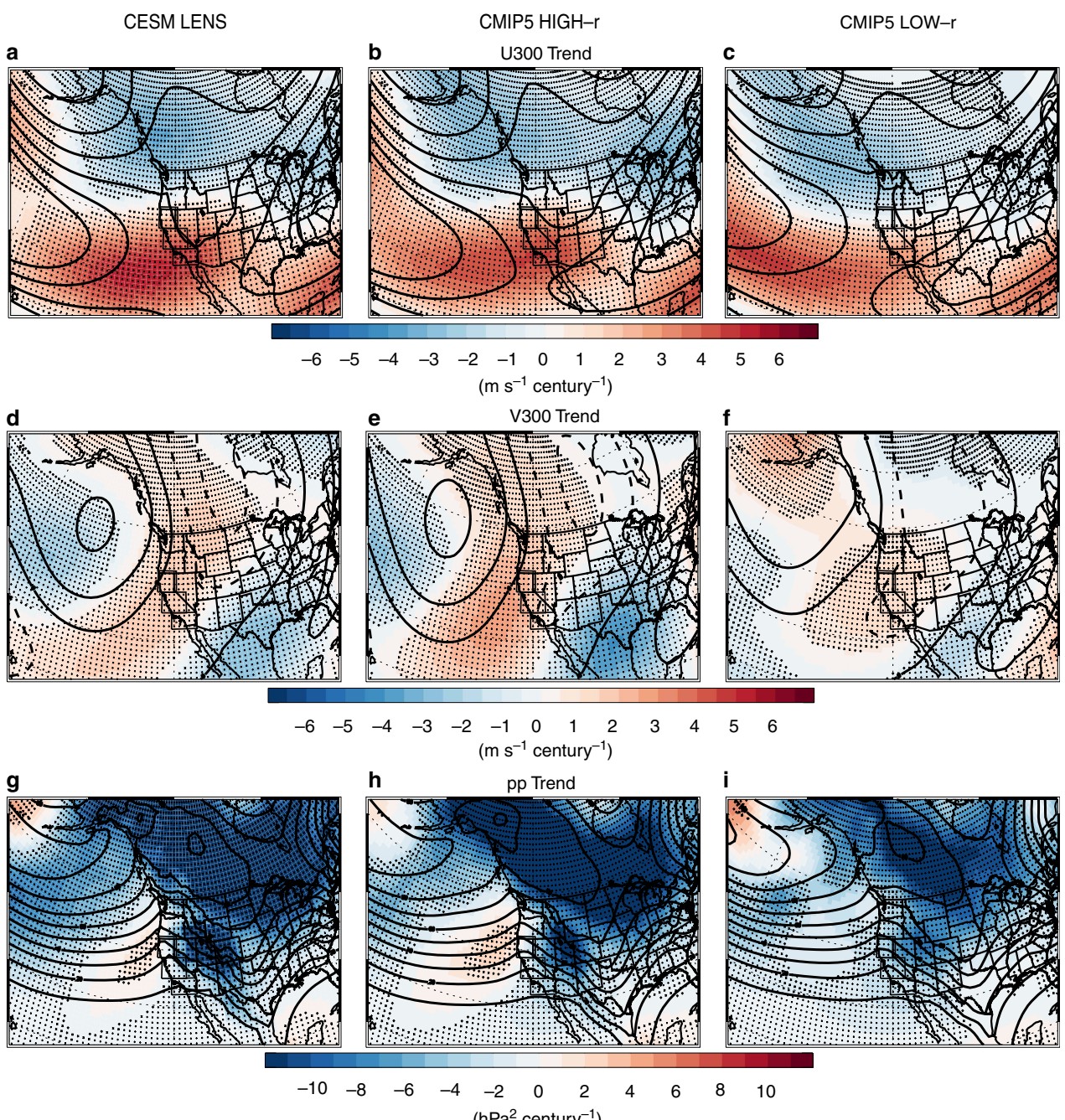

**Figure 4 | 2006–2100 DJF extratropical dynamical changes.** Ensemble mean trend of (**a**–**c**) zonal wind (U300) and (**d**–**f**) meridional wind (V300) at 300 hPa; and (**g**–**i**) storm track activity (*pp*) for (left) the CESM LENS, and the CMIP5 model subset that yields a detrended DJF Niño 3.4 SST versus California precipitation correlation of (centre) at least 0.30 (CMIP5 HIGH-r) and (right) <0.20 (CMIP5 LOW-r). Trend units are m s$^{-1}$ century$^{-1}$ for wind variables and hPa$^2$ century$^{-1}$ for *pp*. Climatological winds are included as thin black contour lines. Contour interval is 5 m s$^{-1}$ (10 hPa$^2$) for wind (*pp*), with negative values indicated by dashed lines. Symbols represent trend significance at the 90% (diamond), 95% (X) or 99% (+) confidence level, accounting for autocorrelation.

the increase in CA precipitation is consistent with an increase in MFC by the mean flow, and more specifically, the component related to dynamical as opposed to thermodynamical processes (that is, changes in wind as opposed to humidity).

Figure 4 shows the DJF CESM LENS ensemble mean trend in zonal (U300) and meridional (V300) wind at 300 hPa. Both show a significant increase over the California region, and more specifically, a southeastward shift of the climatological winds. A similar shift in the zonal winds across CMIP5 models was also found[5], implying an eastward extension of the jet stream, which would help steer mid-latitude storms toward the California coast.

Our results are consistent with this interpretation. The CESM LENS correlation between the U300 trend in the east Pacific and CA precipitation (across realizations) is 0.61, significant at the 99% confidence level (Supplementary Table 3). Moreover, Fig. 4 shows that CMIP5 models that better reproduce the observed correlation between Niño 3.4 SSTs and CA precipitation (that is, CMIP5 HIGH-r) yield a larger southeastward shift of the upper level winds (see also Supplementary Fig. 7). The CMIP5 HIGH-r correlation between the U300 trend in the east Pacific and CA precipitation is 0.72, significant at the 99% confidence level. The corresponding correlation in CMIP5 LOW-r is 0.51, significant at

the 95% confidence level. Similar conclusions exist based on CMIP5 1% $CO_2$ experiments (not shown).

We note another study that relates the increase in precipitation for the west coast of North America to an increase in meridional winds[27]. This, in turn, was related to strengthened zonal mean westerlies in the subtropical upper troposphere (20–40N and 100 hPa), which alters the intermediate-scale stationary wave field. Our analysis suggests the changes in meridional winds are of lesser importance to CA precipitation, as the CESM LENS and CMIP5 LOW-r correlation between the V300 trend in the east Pacific and CA precipitation is not significant at 0.15 and 0.31, respectively. The lone significant correlation occurs in CMIP5 HIGH-r at 0.51 (significant at the 95% confidence level). We also find nonsignificant correlations between CA precipitation and the zonal mean westerlies in the subtropical upper troposphere (20–40N and 100 hPa). The corresponding correlation based on CMIP5 HIGH-r and CMIP5 LOW-r is 0.28 and 0.11, respectively. Furthermore, despite the significant difference in CA precipitation projections between CMIP5 HIGH-r and CMIP5 LOW-r, the ensemble mean trend in 20–40N 100 hPa zonal mean westerlies is similar between the two model subsets, at 8.3 and 8.2 m s$^{-1}$ century$^{-1}$, respectively.

Figure 4g shows the CESM LENS DJF ensemble mean trend in extratropical cyclone (storm track) activity (Methods). In the east Pacific, off the coast of California, a significant increase in storm track activity (pp) exists. The corresponding trend realization agreement shows most grid boxes in the east Pacific have 65–75% of the realizations yielding an increase in pp (significant at the 95% level based on the aforementioned binomial test; Supplementary Fig. 8). Furthermore, a significant correlation between the trend in east Pacific pp and the trend in CA precipitation (across realizations) exists at 0.49 (Supplementary Table 3). Similar results were obtained using CMIP5 models[10]. For our analysis, the corresponding correlation in CMIP5 HIGH-r is 0.81 (significant at the 99% confidence level) and 0.54 (significant at the 95% confidence level) in CMIP5 LOW-r. Figure 4h also shows that CMIP5 HIGH-r yields a larger increase in east Pacific storm track activity relative to CMIP5 LOW-r.

These extratropical DJF changes are consistent with warming of Niño3.4 SSTs and the associated dynamical response in the tropics (that is, the El Niño-like teleconnection; Supplementary Figs 2 and 3). Figures 5 and 6 show that models yield a significant DJF increase in central/eastern tropical Pacific precipitation, weakening of the Walker circulation, a poleward propagating Rossby wave response that originates in the tropical eastern Pacific, and an increase in divergence and Rossby wave generation in the tropical central/eastern Pacific. These features are generally larger, and more robust, in CMIP5 HIGH-r models, including CESM LENS (Fig. 7 and Supplementary Fig. 9). Although model biases may influence this projection, including possible overestimation of tropical convection[28], there are relatively fundamental arguments that support these model projections. The Walker circulation is expected to weaken due to thermodynamical constraints-tropical precipitation increases at a slower rate than water vapour (which increases according to Clausius Clapeyron scaling, assuming constant relative humidity), so the tropical overturning circulation, including the equatorial easterly trade winds, slow down[29]. The Bjerknes feedback, a positive feedback between trade wind intensity and the zonal SST gradient, implies that the above changes would lead to a reduced zonal SST gradient. Warming of the central/eastern tropical Pacific SSTs, in turn, leads to an eastward shift in convection, precipitation and divergence, which would excite a Rossby wave response. Thus, models suggest continued GHG-induced warming will lead to an El-Niño-like climate state in the tropical Pacific[23]. However, as will be discussed below, the increase in CA

precipitation and the associated dynamical responses are not dependent on a reduction in the zonal SST gradient in the tropical Pacific. Model simulations with uniform SST warming also yield a robust increase in CA precipitation, including similar dynamical responses. Moreover, idealized simulations suggest warming of the tropical Pacific-not just the Niño 3.4 SSTs-results in a similar dynamical response and an increase in CA precipitation.

**CMIP5 atmosphere-only simulations.** We also analyse 11 CMIP5 atmosphere model intercomparison project (AMIP) simulations (Supplementary Table 1). These include (1) the control simulation (AMIP), which is forced with 1979–2008 observed SSTs and sea-ice concentrations, as well as GHGs; (2) uniform warming (AMIP4K), which is identical to AMIP, but includes a uniform 4K SST anomaly; and (3) patterned SST warming (AMIP Future), which is identical to AMIP, but includes SST anomalies from the composite SST response from the 1% $CO_2$ coupled CMIP3 model experiments at the time of $CO_2$ quadrupling. Taking AMIP4K-AMIP yields the climate response from a spatially uniform 4K SST warming; AMIP Future-AMIP yields the climate response from a patterned SST warming at the time of $CO_2$ quadrupling, which features enhanced warming of the central/eastern tropical Pacific SSTs.

Both experiments yield a robust increase in ensemble mean CA precipitation: 1.48 mm day$^{-1}$ for AMIP Future and 1.30 mm day$^{-1}$ for AMIP4K, both significant at the 99% confidence level (Fig. 8). This suggests a reduction in the tropical Pacific SST gradient is not necessary for the increase in CA precipitation. AMIP4K and AMIP Future models that were included in our CMIP5 LOW-r subset, including HadGEM2-A and bcc-csm1-1, yield much smaller increases in CA precipitation relative to those models there were included in our CMIP5 HIGH-r subset (including MRI-CGCM3, CCSM4, CanAM4, IPSL-CM5A-LR, IPSL-CM5B-LR and MIROC5). Although this is a small sample size, the ensemble mean CA DJF precipitation response for AMIP4K and AMIP future based on the two models with a lower ability to simulate the observed El Niño teleconnection is 0.34 and 0.27 mm day$^{-1}$, respectively, neither of which are significant. Also note that the precipitation response in these two models is similar to that based on CMIP5 LOW-r: an increase off the Pacific Coast, that does not penetrate into the continental interior, including CA (Supplementary Fig. 10). The corresponding AMIP4K and AMIP Future DJF CA precipitation response based on the six models with a better ability to simulate the observed El Niño teleconnection is 1.26 and 1.81 mm day$^{-1}$, respectively, both significant at the 99% confidence level.

Furthermore, AMIP4K and AMIP Future HIGH-r models yield the extratropical and tropical dynamical responses found in CMIP5 RCP8.5 simulations (Supplementary Figs 11–15). The main difference between AMIP4K and AMIP Future appears to be the ensemble mean tropical precipitation response[30]. The bulk of the precipitation increase in AMIP4K occurs in the South Pacific Convergence Zone. However, some AMIP4K models do yield a maximum precipitation increase in the tropical central/eastern Pacific, and nearly all AMIP4K models yield an (weak) increase in central/eastern tropical Pacific precipitation (Supplementary Fig. 16). Nonetheless, both AMIP4K and AMIP Future yield weakening of the Walker Circulation, an increase in central/eastern tropical Pacific divergence and Rossby wave generation, a poleward propagating Rossby wave, an increase in east Pacific storm track activity (pp), and a southeastward shift of the upper level winds off the CA coast.

Similar to our CMIP5 and CESM LENS analysis, the DJF CA precipitation response across AMIP4K and AMIP Future

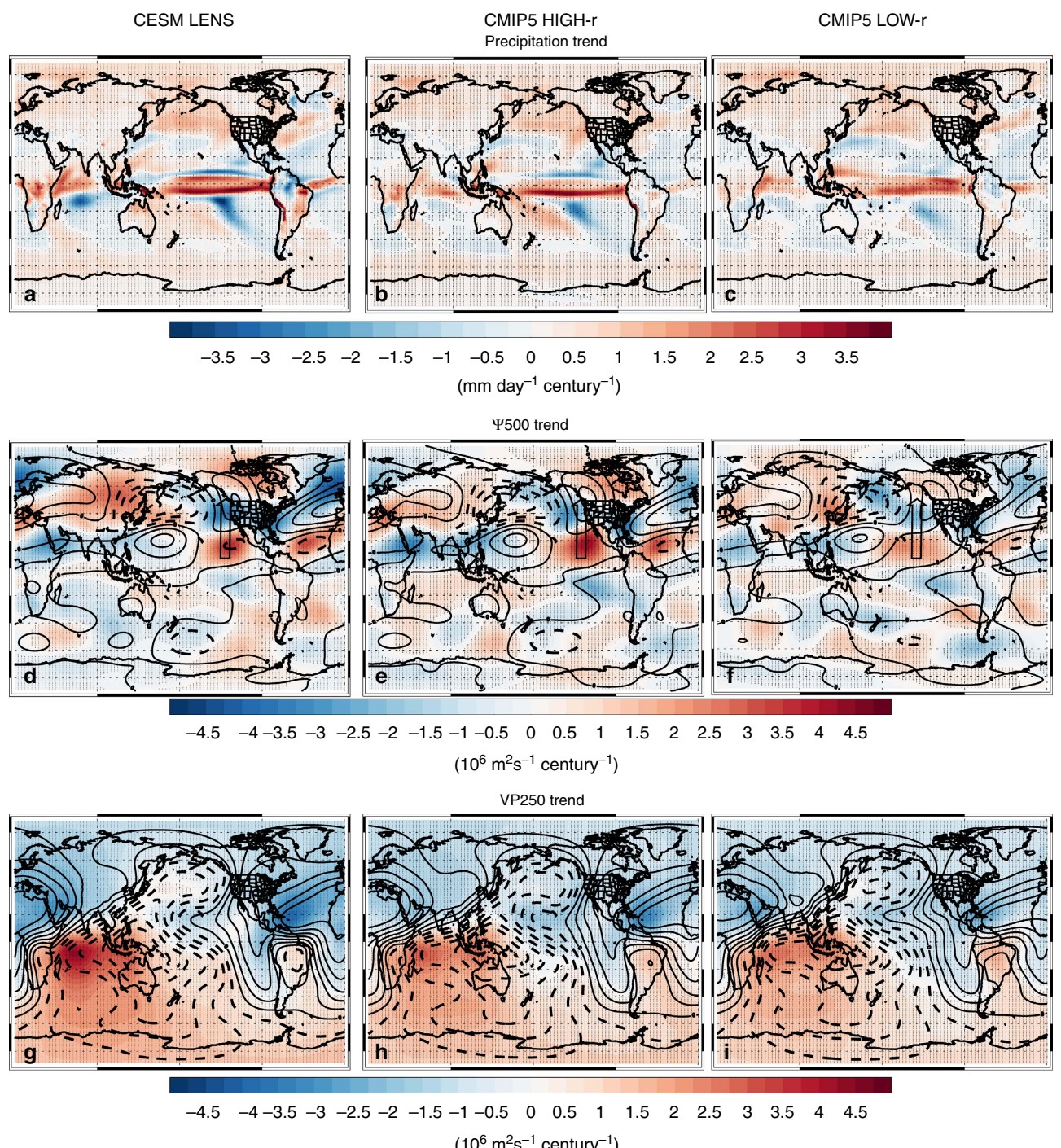

**Figure 5 | 2006–2100 DJF tropical dynamical changes.** Ensemble mean trend of (**a–c**) precipitation; (**d–f**) stationary eddy stream function at 500 hPa ($\Psi$500) and (**g–i**) velocity potential at 250 hPa (VP250) for (left) the CESM LENS, and the CMIP5 model subset that yields a detrended DJF Niño 3.4 SST versus California precipitation correlation of (centre) at least 0.30 (CMIP5 HIGH-r) and (right) <0.20 (CMIP5 LOW-r). Trend units are mm day$^{-1}$ century$^{-1}$ for precipitation and $10^6$ m$^2$ s$^{-1}$ century$^{-1}$ for $\Psi$500 and VP250. Climatological $\Psi$500 and VP250 are included as thin black contour lines. Contour interval is 5 (1) $10^6$ m$^2$ s$^{-1}$ for $\Psi$500 (VP250), with negative values indicated by dashed lines. Black arrows sketch the direction in which the Rossby wave propagates in the Northern Hemisphere. Symbols represent trend significance at the 90% (diamond), 95% (X) or 99% (+) confidence level, accounting for autocorrelation.

models significantly correlates with the east Pacific *pp* and U300 response. The DJF CA precipitation versus *pp* response correlation is 0.93 in AMIP4K and 0.72 in AMIP Future, both significant at the 99% confidence level. The corresponding correlation between DJF CA precipitation and east Pacific U300 is 0.85 in AMIP4K and 0.84 in AMIP Future, both significant at the 99% confidence level. In contrast, the correlation between DJF

CA precipitation and east Pacific V300 is negative at $-0.47$ for AMIP4K and $-0.34$ for AMIP Future, although neither are significant at the 90% confidence level.

We also note that the increase in zonal mean westerlies in the subtropical upper troposphere (20–40N and 100 hPa) does not significantly correlate with the increase in CA precipitation across AMIP models (similar to our results based on CMIP5 models).

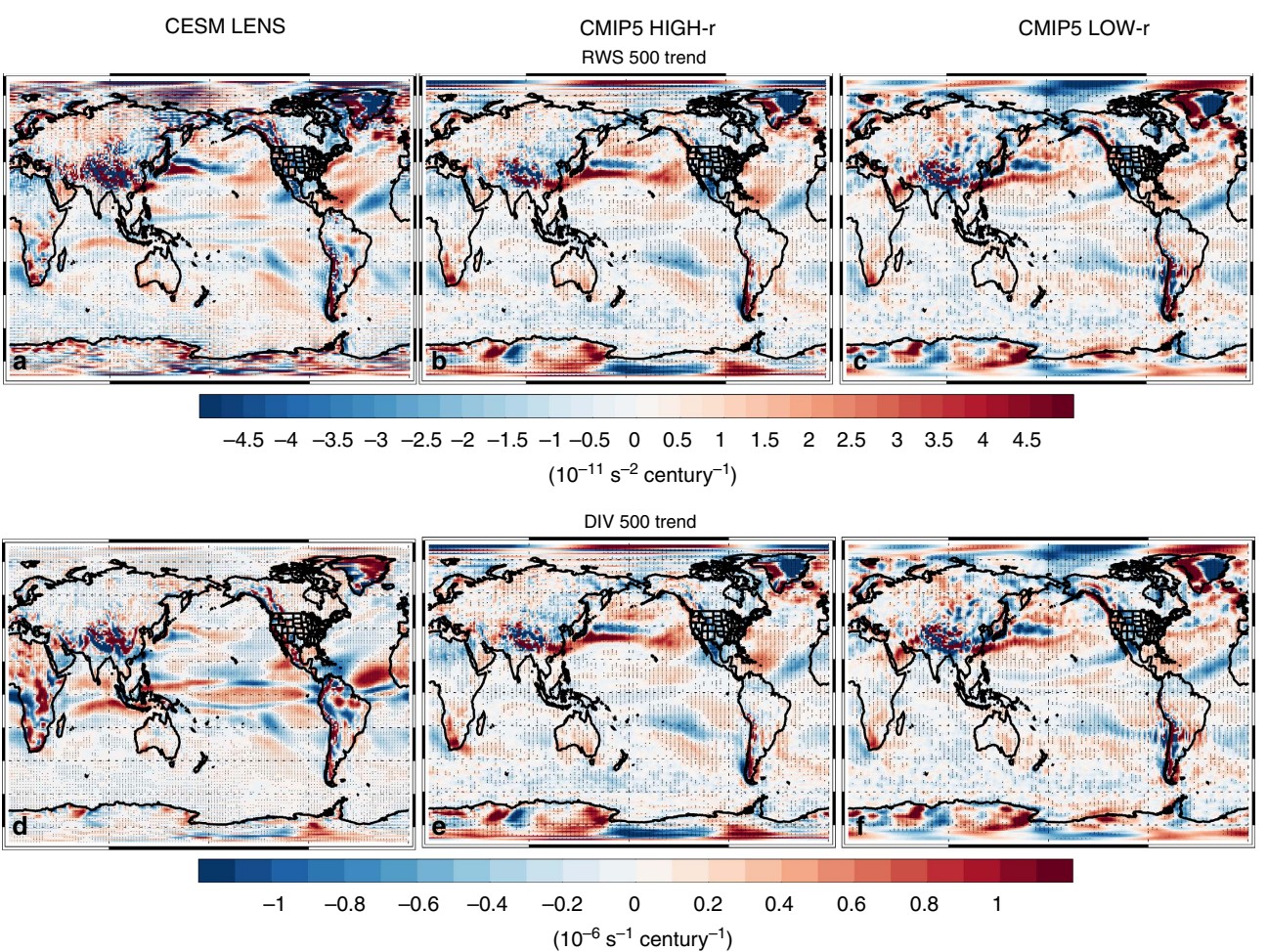

**Figure 6 | 2006–2100 DJF tropical dynamical changes.** Ensemble mean 500 hPa trend of (**a**–**c**) RWS and (**d**–**f**) divergence (DIV) for (left) the CESM LENS, and the CMIP version 5 (CMIP5) model subset that yields a detrended DJF Niño 3.4 SST versus California precipitation correlation of (centre) at least 0.30 (CMIP5 HIGH-r) and (right) <0.20 (CMIP5 LOW-r). Trend units are $10^{-11}\,s^{-2}\,century^{-1}$ for RWS 500 and $10^{-6}\,s^{-1}\,century^{-1}$ for DIV 500. Symbols represent trend significance at the 90% (diamond), 95% (X) or 99% (+) confidence level, accounting for autocorrelation.

In AMIP4K, the corresponding correlation is 0.09; in AMIP Future the correlation is $-0.24$. Thus, our results suggest wetting of CA does not appear to be similar to the relationship between wetting for the west coast of North America and an increase in meridional winds, driven by strengthened zonal mean westerlies in the subtropical upper troposphere[27].

**Connection with tropical pacific sea surface temperatures.** To explicitly show the importance of tropical Pacific SST warming to an increase in CA precipitation during the twenty-first century, we conduct uniform SST warming and present day time slice simulations with CAM5, the atmosphere component of CESM (Methods). CAM5 (and GFDL) models are chosen because they are able simulate the observed correlation between Niño 3.4 SSTs and CA precipitation. However, these models tend to overestimate the sensitivity of CA precipitation to Niño 3.4 SSTs, as well as the CA precipitation climatologies (Supplementary Table 2). Figure 9 shows the CAM5 uniform warming response for DJF precipitation, upper level winds and storm track activity. Similar to AMIP4K simulations, CAM5 shows an increase in CA precipitation; the magnitude of the increase is $1.34\,mm\,day^{-1}$, significant at the 95% confidence level. Furthermore, Fig. 9 show that a similar extratropical dynamical response also occurs, including a southeastward shift of the upper level winds, and an

increase in storm track activity in the east Pacific. The tropical dynamical response also features an increase in central/eastern tropical Pacific divergence, and a poleward propagating Rossby wave that emanates out of the tropical eastern Pacific (Supplementary Fig. 17).

In contrast to the uniform warming response, the climate response without warming of tropical Pacific SSTs yields a muted signal. The hypothetical situation of the tropical Pacific SST not warming as the rest of the globe is a technique to partition regional effects of the overall warming in different locations. Here, CA precipitation decreases by $-0.27\,mm\,day^{-1}$, not significant at the 90% confidence level. Consistently, the dynamical response is also much weaker (Fig. 9). In fact, the east Pacific jet stream and precipitation response moves northward. Similar results (Fig. 10) are obtained with an alternative climate model, GFDL AM3, where CA precipitation increases by $0.92\,mm\,day^{-1}$ (significant at the 95% confidence level) in the uniform warming experiment. Without warming of tropical Pacific SSTs, CA precipitation decreases by $-0.14\,mm\,day^{-1}$, not significant at the 90% confidence level. As with CAM5, the east Pacific jet stream and precipitation response moves northward when tropical Pacific SSTs are not allowed to warm.

The climate response to warming of tropical Pacific SSTs alone is qualitatively similar to, but much stronger than, the uniform

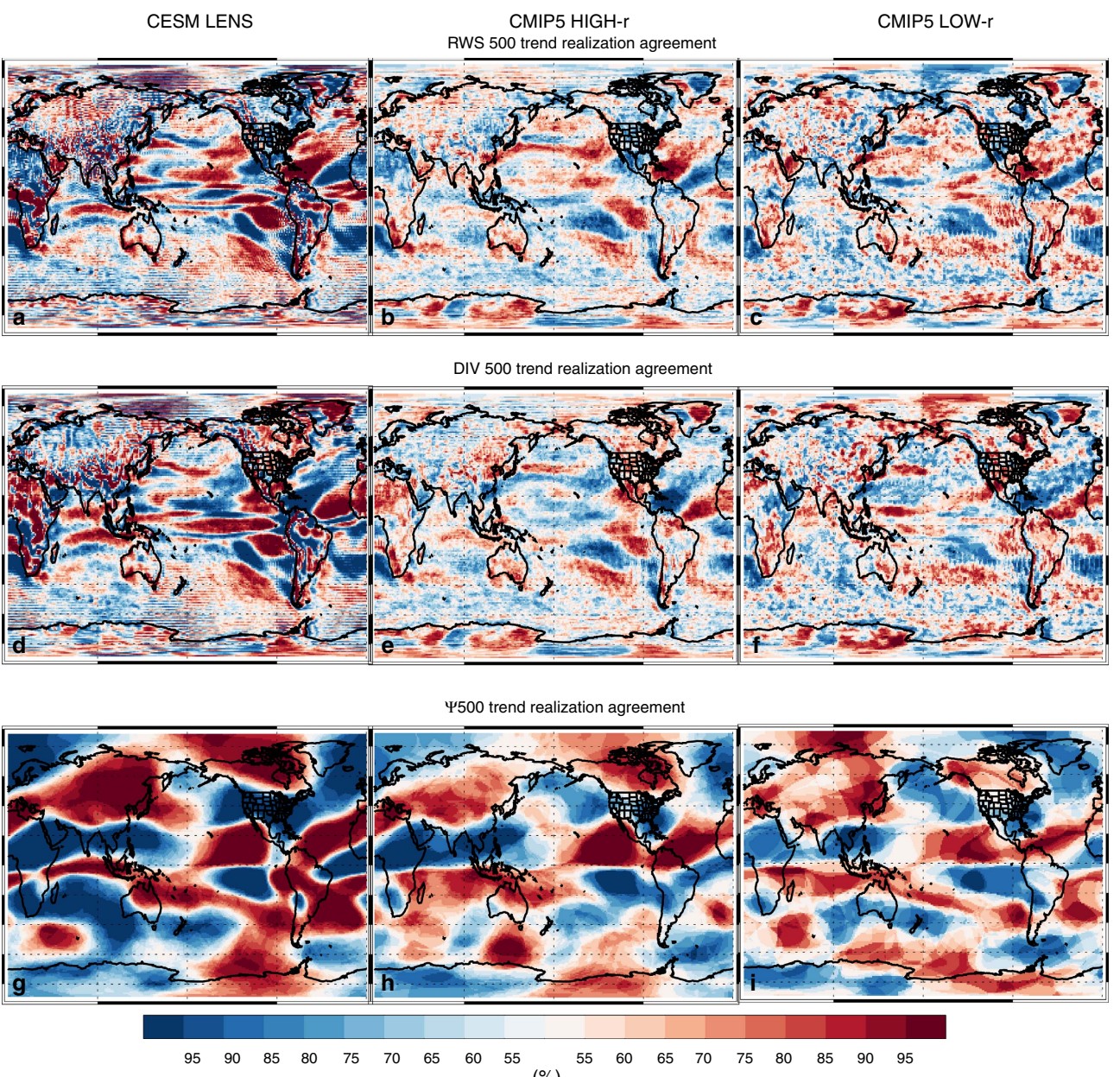

**Figure 7 | 2006–2100 DJF trend realization agreement.** Trend realization agreement (%) for (**a–c**) RWS 500; (**d–f**) divergence (DIV 500); and (**g–i**) stationary eddy stream function (Ψ 500) at 500 hPa for (left) the CESM LENS, and the CMIP version 5 (CMIP5) model subset that yields a detrended DJF Niño 3.4 SST versus California precipitation correlation of (centre) at least 0.30 (CMIP5 HIGH-r) and (right) <0.20 (CMIP5 LOW-r). Warm (cold) colours show the per cent of realizations that yield an increase (decrease) in each quantity.

warming response (Figs 9 and 10). Here, CA precipitation increases by 5.01 mm day$^{-1}$; in GFDL AM3, the corresponding increase is 2.96 mm day$^{-1}$. Both responses are significant at the 99% confidence level. Moreover, the tropical and extratropical dynamical response is very similar to what occurs under uniform warming, including a southeastward shift of the upper level winds and an increase in storm track activity in the east Pacific, as well as an increase in central/eastern tropical Pacific divergence and a poleward propagating Rossby wave response (Supplementary Figs 17 and 18). Similar results are obtained if we vary the location of the uniform tropical SST warming (Methods). Thus, these idealized simulations show the importance of tropical Pacific SST warming to future increases in CA precipitation. Together with the AMIP simulations, our results imply that the increase in CA precipitation may not be dependent on the exact

pattern of SST warming, as long as tropical Pacific SSTs warm-a robust response to future increases in GHGs.

## Discussion

We have shown that there is a dynamically coherent extratropical response to global warming, consistent with previous analyses, involving a southeastward shift of the upper level winds, an increase in storm track activity in the east Pacific, and an increase in CA moisture convergence. We have also shown that there is a dynamically coherent tropical response, including an increase in central/eastern tropical Pacific divergence and a poleward propagating Rossby wave. Both are reminiscent of an El Niño-like teleconnection. CMIP5 models with higher correlation between CA precipitation and El Niño interannual variability yield larger and

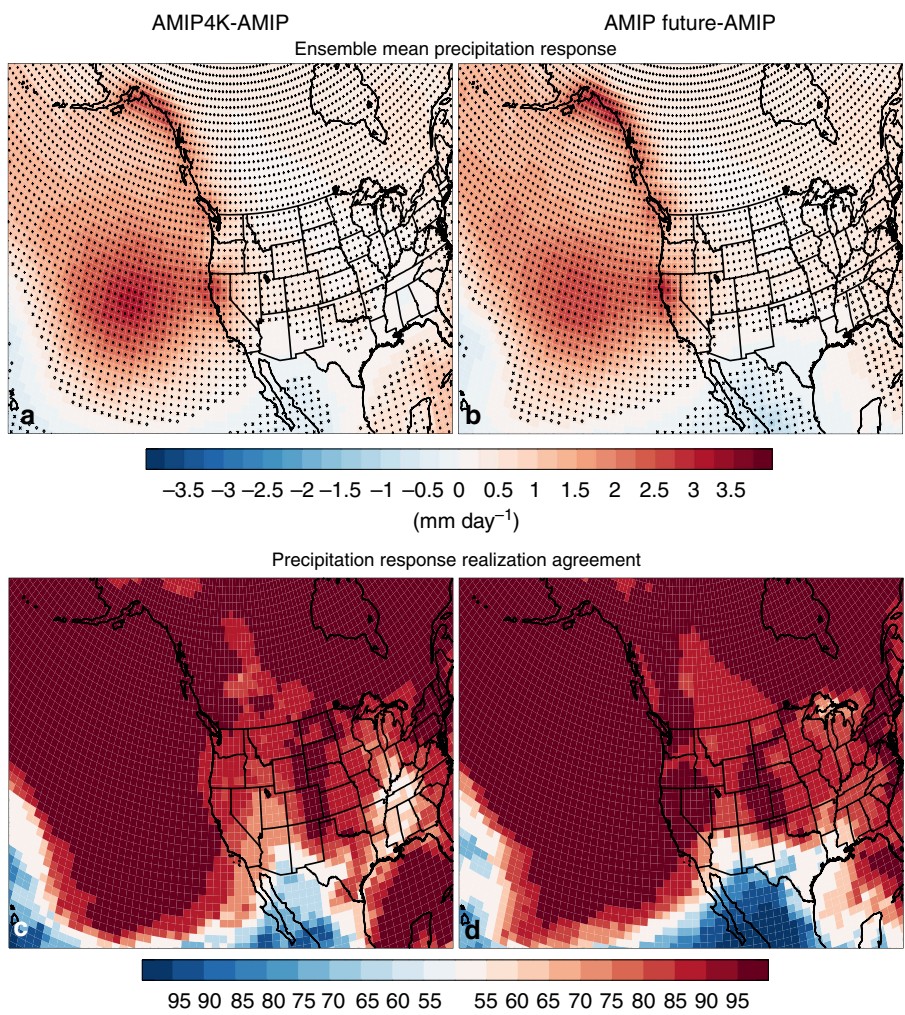

**Figure 8 | California precipitation response in atmosphere-only uniform and patterned SST warming simulations.** (**a,b**) Ensemble mean (left) uniform SST warming (AMIP4K-AMIP) and (right) patterned SST warming (AMIP Future-AMIP) precipitation response (mm day$^{-1}$). (**c,d**) shows the precipitation response realization agreement (%). Symbols in **a,b** represent significance at the 90% (diamond), 95% (X) or 99% ( + ) confidence level, based on a *t*-test for the difference of means.

more consistent increases in California precipitation under global warming. Similar conclusions are obtained when additional criteria are used to subsample the models, including the ability to simulate CA precipitation climatologies, and the observed sensitivity of CA precipitation to El Niño variations.

The increase in CA precipitation and the dynamical responses, however, occur in both patterned and uniform SST warming experiments. This implies El Niño-like tropical Pacific SST warming—a feature of most coupled ocean–atmosphere warming simulations—is not necessary for the response to occur. Using idealized simulations from two different climate models, we show the increase in CA precipitation, and the associated dynamical responses, are associated with SST warming over the entire tropical Pacific. In these models, uniform tropical Pacific warming results in an El-Niño-like teleconnection, and an increase in CA precipitation. We suggest that such a response is related to the zonal heterogeneity of the climatological state of the tropical Pacific-relative to the western tropical Pacific, the central/eastern tropical Pacific features colder SSTs. Since tropical tropospheric temperatures approximately follow a moist-adiabatic temperature profile, tropospheric temperatures in the central/eastern tropical Pacific are also colder than those in the west (not shown). Thus, uniform (and patterned) SST warming of the tropical Pacific results in

greater destabilization of the atmosphere in the central/eastern tropical Pacific, which, in turn, leads to the increase in divergence and Rossby wave response that is associated with the increase in CA precipitation. Although models possess uncertainties, including possible overestimation of tropical convection, our results suggest future GHG-induced warming may lead to an increase in CA precipitation.

## Methods

**Community earth system model large ensemble.** CESM LENS simulations were downloaded from the Earth System Grid at the National Center for Atmospheric Research. They are run at 0.9° × 1.25° resolution, and span 2006–2100 using the business-as-usual scenario, RCP 8.5. Time-varying forcing includes estimated concentrations of GHGs, ozone, and aerosols. Each of the 40 realizations is identically forced; the only difference is the initial condition, which allows the assessment of internal climate variability.

**Rossby wave source.** The Rossby wave source (RWS), which is used to quantify the origin of Rossby waves, is defined as[30,31]

$$RWS = -\zeta D - \mathbf{V}_\chi \bullet \nabla\zeta, \tag{1}$$

where $\mathbf{V}_\chi$ is the divergent wind, $\zeta$ is the vertical component of absolute vorticity and $D$ is the divergence. Monthly data are used to calculate *RWS*, and we have verified that similar results are obtained using limited subsets of daily data (including Reanalysis, CESM LENS and CMIP5). Supplementary Fig. 2 shows that El Niño

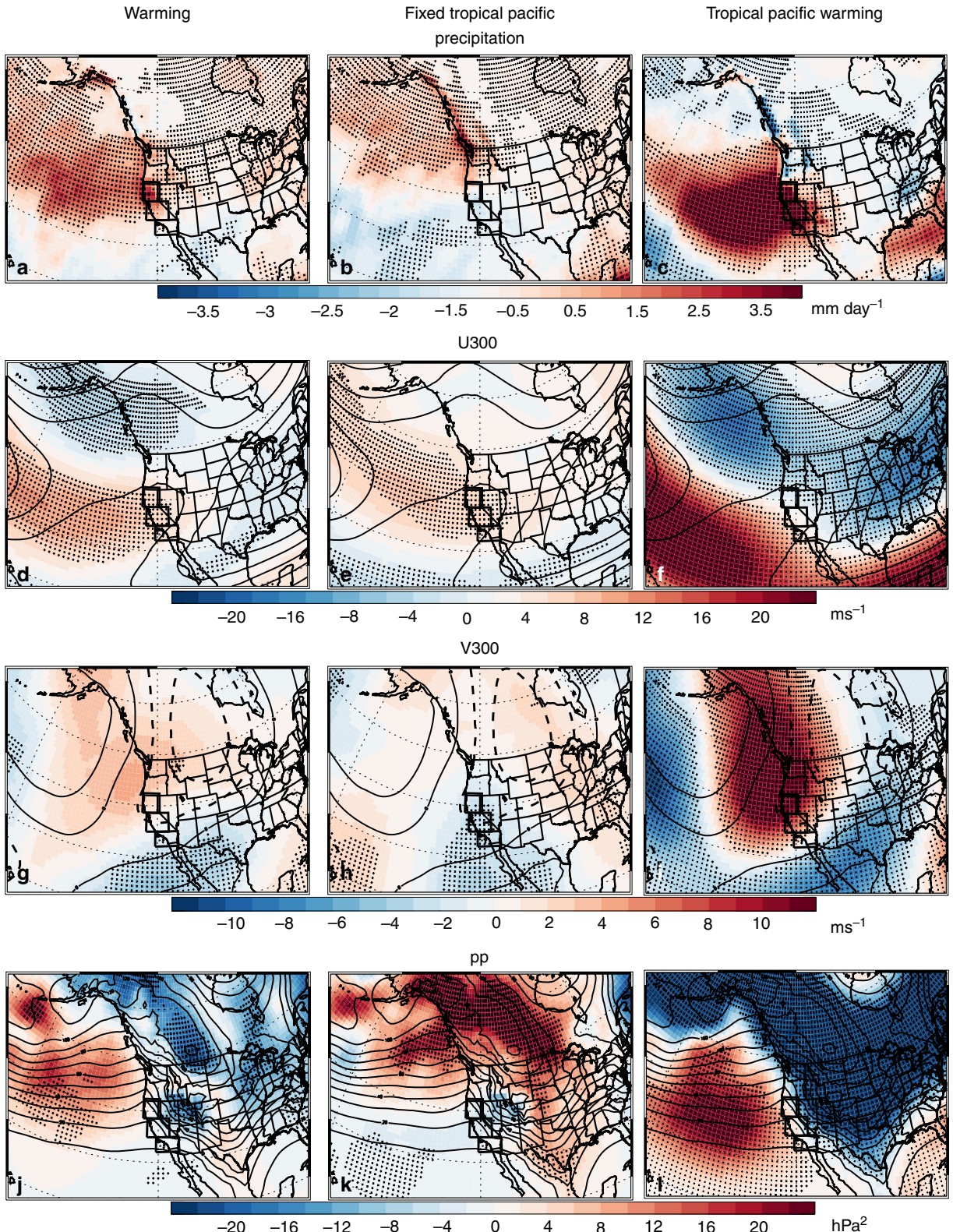

**Figure 9 | Community Atmosphere Model version 5 DJF idealized uniform warming responses.** CAM5 (left) uniform SST warming; (centre) uniform warming with fixed tropical Pacific SSTs; and (right) tropical Pacific SST warming responses for (**a–c**) precipitation (mm day$^{-1}$); (**d–f**) U300 (m s$^{-1}$); (**g–i**) V300 (m s$^{-1}$); and (**j–l**) $pp$ (hPa$^2$). Symbols represent significance at the 90% (diamond), 95% (X) or 99% (+) confidence level, based on a $t$-test for the difference of means. Climatological U300, V300 and $pp$ values are also included as thin black contour lines.

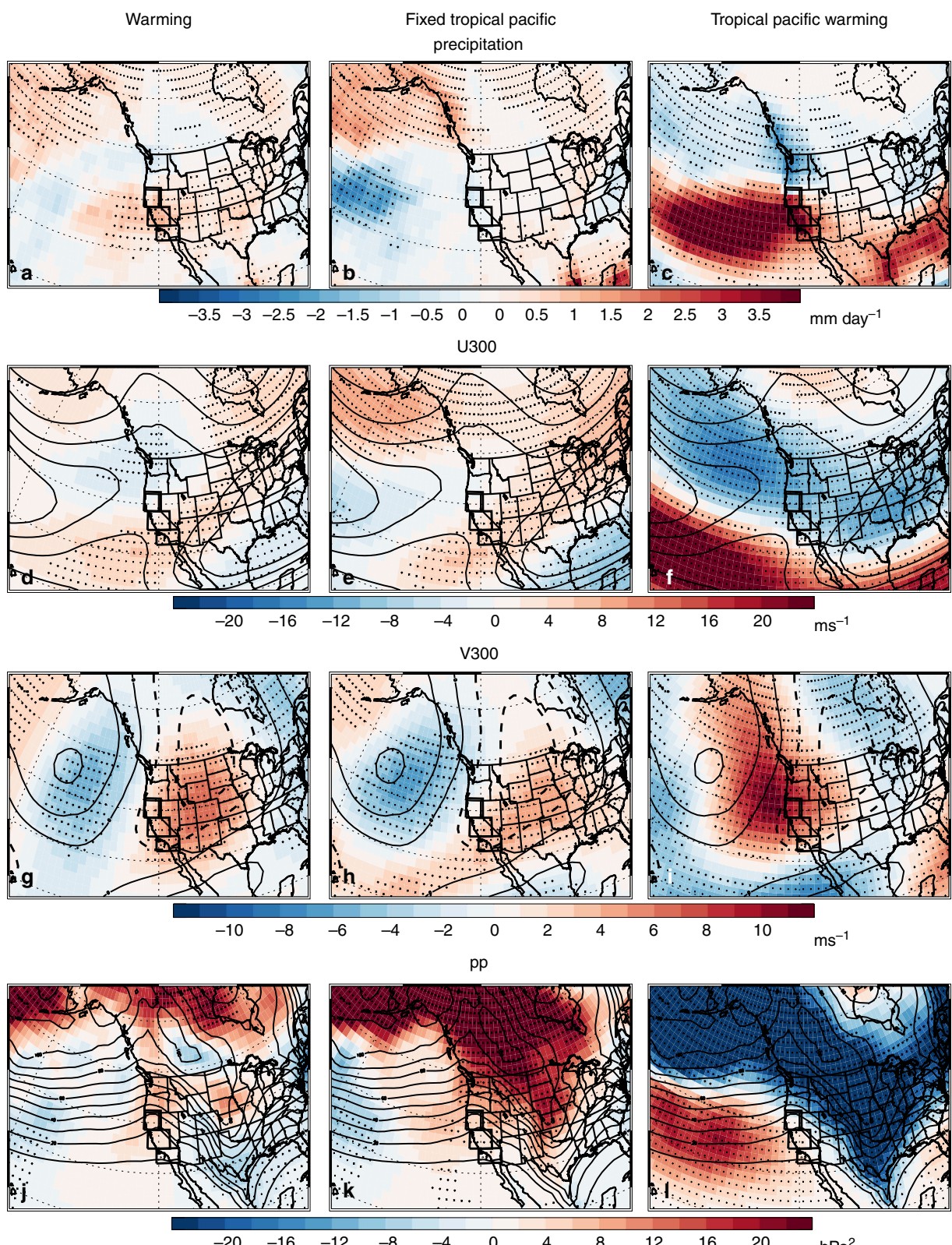

**Figure 10 | Geophysical Fluid Dynamics Laboratory Atmosphere Model version 3 DJF idealized uniform warming responses.** GFDL (left) uniform SST warming; (centre) uniform warming with fixed tropical Pacific SSTs; and (right) tropical Pacific SST warming responses for (**a**–**c**) precipitation (mm day$^{-1}$); (**d**–**f**) U300 (m s$^{-1}$); (**g**–**i**) V300 (m s$^{-1}$); and (**j**–**l**) *pp* (hPa$^2$). Symbols represent significance at the 90% (diamond), 95% (X) or 99% (+) confidence level, based on a *t*-test for the difference of means. Climatological U300, V300 and *pp* values are also included as thin black contour lines.

(that is, warming of Niño 3.4 SSTs) is associated with an increase in Rossby wave generation at 500 hPa near the Niño 3.4 SST region. CMIP5 HIGH-r models reproduce this relationship (as does CESM LENS, not shown). We have also verified that the *RWS* response-including that in the central/eastern tropical Pacific-is dominated by the first term ($-\zeta D$), the generation of vorticity by divergence. Thus, changes in RWS in our experiments are primarily driven by changes in divergence.

**Moisture flux convergence.** The vertically integrated moisture flux **Q** is defined as

$$\mathbf{Q} = \int_0^{p_s} \frac{q\mathbf{V}}{g} \, dp, \tag{2}$$

where $p_s$ is surface pressure, $p$ is pressure, $q$ is specific humidity and **V** is the horizontal wind vector (vectors are denoted by bold font). The zonal and meridional component of the vertically integrated moisture flux, $Qu$ and $Qv$, are written as

$$Qu = \int_0^{p_s} \frac{qu}{g} \, dp; \quad Qv = \int_0^{p_s} \frac{qv}{g} \, dp, \tag{3}$$

At monthly timescales ($> 10$ days), the atmospheric moisture budget indicates that precipitation, evaporation and net moisture flux into/out of an atmospheric column through its lateral boundary are in a balance[25]. The atmospheric moisture balance is written as

$$p = -\nabla \bullet \mathbf{Q} + e. \tag{4}$$

where $p$ is precipitation, $e$ is surface evaporation and $-\nabla \bullet \mathbf{Q}$ is the vertically integrated moisture flux convergence.

The areal-averaged moisture flux can be estimated by direct calculation of the convergence field of moisture flux over a region, or by calculation of the net moisture flux into/out of the regional lateral boundary, which is subsequently divided by the area of the region[32–34]. We use the latter approach, and divide California into three sub-regions that represent southern (32.0–34.9°N; 239.4–245.6°E), central (34.9–38.6°N; 236.9–243.1°E) and northern (38.8–42.4°N; 235.6–240.6°E) California.

The zonal and meridional moisture flux ($Fu$ and $Fv$) crossing the boundary of each sub-region is calculated by[32]

$$Fu = \int Qu Re d\phi; \quad Fv = \int Qv Re \cos\phi d\lambda, \tag{5}$$

where $Re$, $\phi$ and $\lambda$ are the radius of the Earth, latitude and longitude, respectively. The area integration of the vertically integrated MFC is obtained by

$$MFC = \int -\nabla \bullet \mathbf{Q} dS = \oint -\mathbf{Q}_n ds = \sum (Fu \ and \ Fv) = Fi - Fo \tag{6}$$

where $dS$, $ds$, $Fo$ and $Fi$ are the area element, the line element along the boundary of the relevant subregion, and moisture outflow and moisture inflow, respectively. $\mathbf{Q}_n$ is the normal component of **Q** to $ds$. The area integrated precipitation $P$ and evaporation $E$ are defined as

$$P = \int p dS; \quad E = \int e dS. \tag{7}$$

The area element is given by $dS = Re^2 \cos\phi d\phi d\lambda$. The moisture balance equation for each subregion is given by

$$P = MFC + E = Fi - Fo + E. \tag{8}$$

**Decomposition of moisture flux convergence.** We define an overbear to be a monthly time average and a prime to denote the departure from that mean (that is, $u = \bar{u} + u'$). Covariances, such as those between zonal wind $u$ and specific humidity $q$, which give the zonal moisture flux, can be written as $\overline{uq} = \bar{u}\bar{q} + \overline{u'q'}$. Thus, when time averages (that is, monthly means) are taken, the moisture flux can be divided into mean and transient components. The CESM LENS archive includes the monthly mean components, and well as the monthly mean covariances (for example, $\overline{uq}$). The transient component is therefore estimated as: $\overline{uq} - \bar{u}\bar{q}$. This allows calculation of the corresponding mean and transient MFC.

The mean MFC is determined by both specific humidity and wind velocity. These two variables are associated with thermodynamic and dynamic aspects of the atmosphere, respectively[26,33,34]. To investigate the relative roles of thermodynamic versus dynamic processes on the mean MFC, we express the specific humidity and wind velocity for each month as $\bar{q} = \bar{q}_c + \bar{q}_a$ and $\bar{\mathbf{V}} = \bar{\mathbf{V}}_c + \bar{\mathbf{V}}_a$, where $\bar{q}_c$ and $\bar{\mathbf{V}}_c$ are the 2006–2100 monthly mean climatology of specific humidity and wind velocity, respectively; $\bar{q}_a$ and $\bar{\mathbf{V}}_a$ are the monthly deviations from the 2006 to 2100 monthly mean climatology. The change in *MFC* can therefore be expressed as the sum of three components: mean MFC associated with thermodynamics (humidity), mean MFC associated with dynamics (wind), and the transient MFC (that is, the highly variable, day-to-day weather).

The change (temporal variation) in the mean vertically integrated moisture transport is decomposed into a thermodynamic (THERM) and a dynamic

(DYNM) term[33,34].

$$Qu_{DYNM} = \int_0^{p_s} \frac{\bar{q}_c \bar{u}_a}{g} \, dp; \quad Qv_{DYNM} = \int_0^{p_s} \frac{\bar{q}_c \bar{v}_a}{g} \, dp, \tag{9}$$

$$Qu_{THERM} = \int_0^{p_s} \frac{\bar{q}_a \bar{u}_c}{g} \, dp; \quad Qv_{THERM} = \int_0^{p_s} \frac{\bar{q}_a \bar{v}_c}{g} \, dp. \tag{10}$$

This allows calculation of the corresponding mean MFC, due to thermodynamic versus dynamic controls.

**Storm track activity.** We define extratropical cyclone (storm track) activity based on temporal variance statistics, band-pass filtered using a 24 h difference filter[10]

$$pp = \overline{[slp(t + 24 \, hr) - slp(t)]^2}, \tag{11}$$

where $slp$ is sea level pressure and $pp$ is the 24 h difference filtered variance of sea level pressure. The overbear corresponds to time averaging over the winter (DJF) season. Because limited sub-daily data is available, we use daily data. We note that similar $pp$ results are obtained from reanalysis data using 6-hourly or -daily data (not shown).

California DJF precipitation is significantly correlated with DJF storm track activity off the coast of California. The 1979/80 to 2014/15 correlation between east Pacific $pp$ from National Center for Environmental Prediction/National Center for Atmospheric Research reanalysis and CA precipitation from Global Precipitation Climatology Project is $\sim 0.80$. Here, the 'east Pacific' is defined as the area covering the region of highest CA precipitation-$pp$ correlation. We adopt a similar definition of the east Pacific, but based on the ensemble mean CESM LENS correlation between $pp$ and CA precipitation. The corresponding, ensemble mean (detrended) correlation between east Pacific $pp$ and CA precipitation is 0.85 (Supplementary Table 3). Thus, as with most CMIP5 models[10], CESM LENS is able to reproduce this relationship.

**Idealized simulations.** Uniform SST warming CAM5 simulations-the atmospheric component of CESM-are conducted at $0.9° \times 1.25°$ resolution, and integrated for 10 years. This experiment features SSTs that are 3.3 K warmer than the control, which corresponds to the annual mean SST warming anomaly (2090–2099 relative to 2006–2015) from the ensemble mean of the 40 CESM LENS realizations. We also conduct an additional uniform warming experiment that is analogous to the original, but tropical Pacific SSTs (5S–5N; 120-270E) are held constant. Finally, we conduct a uniform warming experiment where only tropical Pacific SSTs are increased (by 3.3 K). Similar experiments are conducted with GFDL AM3. Here, the SST anomaly is 3.0 K, which corresponds to the annual mean SST warming anomaly from the RCP8.5 ensemble mean. Like CAM5, GFDL AM3 is able simulate the observed teleconnection between Niño 3.4 SSTs and CA precipitation (Supplementary Table 1). However, these models tend to overestimate the sensitivity of CA precipitation to Niño 3.4 SSTs (Supplementary Table 2), and also tend to overestimate CA precipitation climatologies-especially in the case of GFDL.

We also conduct similar uniform warming experiments (not shown) that focus on the Niño 3.4 SST region (as opposed to the entire tropical Pacific), including simulations that lack Niño 3.4 SST warming, or include only Niño 3.4 SST warming. These Niño 3.4 SST experiments yield qualitatively similar results as those based on uniform tropical Pacific SST warming-without warming of Niño 3.4 SSTs, the increase in CA precipitation, and the dynamical responses, are significantly muted.

Finally, we conduct additional uniform tropical SST warming experiments, including (1) uniform warming of the entire tropics (5S–5N; 0-360E) and (2) uniform warming of the tropical and subtropical Pacific (15S–15N; 120-270E). Both experiments yield results consistent with tropical Pacific (5S–5N; 120-170E) warming, including similar dynamical responses and a significant increase in CA precipitation. For example, uniform tropical warming yields a CA DJF precipitation increase of 1.81 (2.44) mm day$^{-1}$ in CAM5 (GFDL). Uniform tropical and subtropical Pacific warming yields a CA DJF precipitation increase of 9.82 (6.37) mm day$^{-1}$ in CAM5 (GFDL). Thus, we find an increase in CA precipitation is a robust feature to several patterns of uniform tropical warming.

We also conduct patterned SST warming experiments for both CAM5 and GFDL, which also include decreases in sea-ice and increases in GHGs (not shown). For CAM5, SST and sea-ice concentration anomalies (2090–2099 relative to 2006–2015) from the ensemble mean of the 40 CESM LENS simulations are used. GHG concentrations and aerosol emissions are also set to the RCP8.5 2090–2099 mean. For GFDL, GHG concentrations are set to the RCP8.5 2090–2099 mean, and SST and sea-ice concentration anomalies (2090–2099 relative to 2006–2015) from the CMIP5 ensemble mean are used. These experiments are designed to resemble the CMIP5 RCP8.5 simulations, which include both patterned SST warming and increases in GHGs. We also conduct an additional patterned warming experiment, analogous to the original, but Niño 3.4 SSTs are held constant. Both CAM5 and GFDL patterned SST warming experiments yield a similar response, including a significant increase in DJF CA precipitation (2.04 and 1.66 mm day$^{-1}$, respectively). Furthermore, both models yield tropical and extratropical dynamical responses similar to those in CMIP5 RCP8.5 models. However, a muted signal is obtained when Niño 3.4 SSTs are fixed. CA precipitation increases by only 0.38

(0.20) mm day$^{-1}$, not significant at the 90% confidence level, in CAM5 (GFDL). Consistently, the dynamical response is also much weaker.

**Trend and correlation significance.** Ensemble mean trend significance is based on a standard $t$-test, accounting for the influence of serial correlation by using the effective sample size, $n(1-r_1)(1+r_1)^{-1}$, where $n$ is the number of years and $r_1$ is the lag-1 autocorrelation coefficient. Significance of CAM5, GFDL AM3 and AMIP time slice simulations is based on $t$-test for the difference of means, using the pooled variance. Significance of correlations is also based on a $t$-test with $N-2$ degrees of freedom, with the $t$-statistic equal to $r/[(1-r^2)/(N-2)]^{0.5}$. $N$ is the sample size (for example, number of years) and $r$ is the correlation. Detrended correlations are estimated by first detrending the corresponding time series, and then calculating the correlation.

**Code and Data availability.** The data and code that support the findings of this study are available from the corresponding author upon reasonable request.

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

## Acknowledgements

This study was funded by NASA Earth and Space Science Fellowship NNX15AN12H and NSF award AGS-1455682. We thank D. Neelin.

## Author contributions

R.J.A. conceived the project, designed the study and wrote the paper. R.J.A. carried out the bulk of the analysis, with R.L. assisting with analysis of the CESM LENS experiments.

## Additional information

**Competing interests:** The authors declare no competing financial interests.

