## [Peer Review File · Nature Communications]

Reviewers' comments:

Reviewer #1 (Remarks to the Author):

I appreciate the authors work to address the comments in my second review, in particular they changed the title and added text and figures to clarify their proposed mechanism. However, I require clarification on some key points before I can recommend publication.

1) The fact that the circulation and CA precipitation responses to uniform warming (AMIP4K) and patterned warming (AMIPFuture or the authors experiments of only warming in Nino 3.4 region) are similar calls into question the authors mechanism outlined on lines 177-182. There is no change in the zonal SST gradient in AMIP4K. Thus, the similarity of the responses to CO2 and El Nino cannot rely on SST gradient changes and the Bjerknes effect. In particular, the uniform warming experiment does not represent a shift toward an El Nino like SST state (there is no change in zonal SST gradient) yet the circulation and CA precipitation responses are consistent with warming only in Nino 3.4 region. Changes in vertical stability are not discussed by the authors and they may be more important than SST gradient changes. The authors need to revise the discussion of their mechanism to address the fact that the Rossby wave teleconnection can occur in the absence of a change in zonal SST gradient.

2) In the revised manuscript, the authors discuss how changes in divergence and Rossby wave source (RWS) drive a poleward propagating Rossby wave (Fig. 6). Generating a Rossby wave requires absolute divergence and positive RWS in the upper troposphere (not anomalous divergence and RWS). The authors should confirm there is absolute divergence and positive RWS in the central/east Pacific for the warmed climate. Furthermore, they should identify the poleward propagating Rossby wave more clearly by labelling the associated anticyclonic and cyclonic wave train in their streamfunction figure. In order to properly interpret and compare the RWS, divergence and streamfunction responses please show them all at the same pressure level, e.g. 250 hPa.

Reviewer #2 (Remarks to the Author):

I am generally supportive of publication of this work and wish to encourage the authors' efforts in this important area. However, I do have substantial reservations about the level of certainty with which the conclusions are phrased, and feel that the study would benefit from improvements in the way it is rooted in prior literature on this topic.

This study is a significant contribution in terms of following up on the question raised in reference 5 regarding how changes in the Eastern Pacific jetstream and storm track coming onto California affects California precipitation, especially in light of how the region since the transition between the tendency towards increased precipitation to the north and decreased precipitation to the south. This hypothesis was motivated by the principal mechanism by which El Niño affects California rainfall, so that study used El Niño interannual variability to check the precipitation to Eastern-Pacific jet relations. And because long-term El Niño-like change is one of the most obvious hypotheses for the jet impacts on precipitation change in California, that study also explicitly checked for a correlation to El Niño indices, such as Niño 3.4 SST, across the portion of the CMIP5 ensemble that was available. Niño 3.4 SST per se was not found to be a strong correlation but large-scale changes in the jet occur under global warming independent of equatorial cold tongue SST were found to yield a pathway similar to the El Niño effect of stirring additional storms onto the coast. That study left the exact source of the jetstream changes undetermined, so mechanistic studies such as the ones presented in the current Allen and Luptowitz manuscript can be highly valuable in determining the physical source of these changes. But it would be beneficial to more clearly indicate what has already been examined. [In passing, it's worth noting that there is

substantial parallels in parts of the abstract of the present manuscript and reference 5 that might be worth referencing within the abstract itself since this is permitted in Nature journals].

A subsequent study, Langenbrunner et al. (2015, *J. Climate*) [not currently cited in the manuscript] looked at the CMIP5 model ensemble uncertainty in this region, and also explicitly sought an explanation in tropical Pacific sea surface temperature. It found that the leading mode affecting the inter-model range of projections for California rainfall change under global warming was related to a more complex large-scale pattern of SST changes with little expression in the equatorial cold tongue region. A second mode of North Pacific intermodel precipitation change differences was related to SST patterns that included a substantial El Niño like component, but this had relatively modest expression in California rainfall. Thus the model response that affects both the jet and California rainfall across the CMIP5 ensemble appears to have other contributions than El Niño like changes. It would be appropriate for the present study to frame itself with respect to this prior work; it would then be clear the hypothesis has been previously examined at the basic level of relations across the CMIP5 ensemble and would highlight the question of what aspect of the current CMIP5 analysis suggests an El Niño like contribution can be significant for some part of the model ensemble.

The answer to this question in the present manuscript is conditioning on high correlation between California rainfall and El Niño SST indices in interannual variability. The authors show that an average over a subset of models with high interannual CA-Precip–equatorial-Pacific-SST correlation (“high-r”) tends to exhibit larger California precipitation changes relative to an average over a low-correlation subset. This is a nice analysis and a very reasonable way to parse out the results in a way that distinguishes certain behaviors.

However, high-r alone does not guarantee a more realistic response. Overly high correlation would also be unrealistic and high correlation can go with unrealistic amplitude response. Some models exhibit larger-than-observed regressions of precipitation on Niño indices in this region. This was true of the previous version of the NCAR model (e.g., Langenbrunner et al. 2013), and may well be inherited by the version used here. Furthermore, many models have a storm track that hits the US West Coast too far south. For such models there is considerable concern that increases in California are overestimated, although the increase at the end of the storm track may be qualitatively correct but simply misplaced in latitude along the coast. Such models can easily exhibit excessive rainfall sensitivity in California. If one took a subset of CMIP5 models conditioned on an interannual regression coefficient within the error bars of the observational coefficient, and a condition for a correct climatological precipitation latitude dependence in California within the observational error would this subensemble yield the same results? The high correlation alone is not sufficient to fully support statements such as “models that better simulate the observed El Niño-CA precipitation teleconnection” (lines 13-14), but inclusion of a measure of regression coefficient and climatological precipitation would substantially help.

In the SST experiments, similar concerns arise. Both the GFDL model and the NCAR large ensemble exhibit versions of this issue in the position of the southern boundary of the storm track where it arrives at the US West Coast, tending to extend too far south. This does not invalidate qualitative results using those models here, but it does indicate substantial caveats that should be acknowledged. The experiments in Figure 7 are not convincing because the SST gradients are so much larger than the range of anything that would ever occur: the experiment with no warming in Niño 3.4 SST is similar to having a 3 or 4° La Niña relative to an overall warm climate. And the question to be addressed is not whether A) an enormous El Niño or La Niña can have an impact on California, but whether B) the relatively modest equatorial temperature variations relative to the overall warmer climate adds anything significant to the changes to the jetstream that would occur without these, associated with larger scale warming. Figure 7 shows the former — which is unsurprising and serves as a red herring in terms of the flow of the argument — but does not add to the evidence for the latter. Supplementary figures 11-12 suffer from this same issue. Figure S13 in the supplement shows an experiment that does appear to better address this issue in that it

compares a patterned SST case to a uniform SST warming case. This doesn't distinguish between large-scale tropical-extratropical gradients and local Pacific enhancements, but it does at least distinguish from a uniform warming. The description of these SST experiments could be more clearly written — it takes a while to parse out what is specifically being compared in each case. But I would recommend swapping figure S13 from the supplement into the main text, and moving figure 7 into the supplement. The results that are alluded to in line 218-219 "idealized CAM5 simulations forced with uniform tropical Pacific warming also reproduce these results (not shown)" are very intriguing. They seem to indicate that amplified warming in the equatorial Pacific is not essential, consistent with examination of the equatorial Pacific effect in the earlier studies. It would be worth including these experiments. That would suggest that a broad scale tropical warming could affect the jet stream and that would still be consistent with models with higher El Niño response in California reacting more strongly to this jetstream change.

Turning to the conclusions, in both the abstract line 15 "Our results suggest that California will likely become wetter in a warmer world." and at the end of the discussion line 232 "[these results] imply that California will likely become wetter in response to continued greenhouse gas emissions." These statements seem overly confident in light of the model uncertainties and of the prior analysis discussed above. California precipitation projections are a serious matter, with real consequences for infrastructure decisions and for public confidence in climate science. Both the "likely" and "wetter" are aspects of this phrasing that are open to misinterpretation. While the evidence here is useful in moving the understanding of these projections forward, it elucidates a factor rather than leading to an overall conclusion of likely increases. Furthermore, "wetter" refers only to the precipitation, whereas there is a substantial contribution to the water budget as experienced on the ground due to evapotranspiration increases (e.g. reference 24). This is not addressed here, and yet would be important factor before one could make a sweeping statement about wetter climate.

I'd like to make suggestions on how to revise these conclusions, but first let me turn to some strengths of this paper that deserve highlighting. In the dynamical analysis lines 114-182, the results are very coherent. The findings regarding meridional wind versus zonal wind help to distinguish between two dynamical hypotheses. The authors do a nice job of bringing together storm track activity measures from reference 9 [although they could be a little clearer in attributing these measures] with wind measures and moisture convergence measures to provide a thorough picture of dynamical changes associated with the high-r models.

Revising the conclusions to adhere more closely to what has been specifically shown: there is a dynamically coherent response, consistent with previous analysis, involving a southeastward shift of the upper level winds and an increase in storm track activity in the east Pacific, and an increase in CA moisture convergence and that models with higher correlation between California precipitation and El Niño interannual variability yield larger and more consistent increases in California precipitation under global warming. And that while local enhancement of cold tongue SST does not appear to be essential to the changes in the jetstream and associated, or precipitation changes, there are indications that they can enhance it.

Response to Reviewers

We thank both reviewers for their thorough evaluation of our paper, as well as their helpful suggestions, which have significantly improved our manuscript. Below are our responses to each comment.

Reviewer #1:

I appreciate the authors work to address the comments in my second review, in particular they changed the title and added text and figures to clarify their proposed mechanism. However, I require clarification on some key points before I can recommend publication.

1) The fact that the circulation and CA precipitation responses to uniform warming (AMIP4K) and patterned warming (AMIPFuture or the authors experiments of only warming in Nino 3.4 region) are similar calls into question the authors mechanism outlined on lines 177-182. There is no change in the zonal SST gradient in AMIP4K. Thus, the similarity of the responses to CO₂ and El Nino cannot rely on SST gradient changes and the Bjerknes effect. In particular, the uniform warming experiment does not represent a shift toward an El Nino like SST state (there is no change in zonal SST gradient) yet the circulation and CA precipitation responses are consistent with warming only in Nino 3.4 region. Changes in vertical stability are not discussed by the authors and they may be more important than SST gradient changes. The authors need to revise the discussion of their mechanism to address the fact that the Rossby wave teleconnection can occur in the absence of a change in zonal SST gradient.

We have keep much of this discussion, since it is likely remains relevant for the patterned warming response. We now make it clear, however, that the CA precipitation and dynamical responses occur in both uniform and patterned warming experiments, implying changes in the zonal SST gradient do not appear to be necessary. We suggest warming of the tropical Pacific (not just the Nino3.4 SST region) is able to cause these responses, and show this in new idealized CAM5 and GFDL simulations. We also include a brief discussion on why we suspect uniform tropical Pacific warming leads to an El-Nino like teleconnection, which involves zonal heterogeneity in the tropical Pacific climatological state. Relative to the western tropical Pacific, the central/eastern tropical Pacific features colder SSTs. Since tropical tropospheric temperatures approximately follow a moist-adiabatic temperature profile, tropospheric temperatures in the central/eastern tropical Pacific are also colder than those in the West (which we have verified). Thus, uniform (and patterned) SST warming of the tropical Pacific results in greater destabilization of the atmosphere in the central/eastern tropical Pacific, which, in turn, leads to the increase in divergence and Rossby wave response that is associated with the increase in CA precipitation.

2) In the revised manuscript, the authors discuss how changes in divergence and Rossby wave source (RWS) drive a poleward propagating Rossby wave (Fig. 6). Generating a Rossby wave requires absolute divergence and positive RWS in the upper troposphere (not anomalous divergence and RWS). The authors should confirm there is absolute divergence and positive RWS in the central/east Pacific for the warmed climate. Furthermore, they should identify the poleward propagating Rossby wave more clearly by labelling the associated anticyclonic and cyclonic wave train in their streamfunction figure. In order to properly interpret and compare the

RWS, divergence and streamfunction responses please show them all at the same pressure level, e.g. 250 hPa.

All RWS, divergence and streamfunction plots are now based on the same pressure level. We use a mid-troposphere value—500 hpa—because this is where the response is most robust across all of the simulations we analyze. We have also added arrows to the streamfunction plots to more clearly show the poleward propagating Rossby wave response in the Northern Hemisphere. We also confirm that there is absolute divergence and positive RWS in the central/eastern Pacific in the warmed climate.

The figure below shows DIV 500 over the latter half of the 21st century in CMIP5 HIGH-r models.

The figure below shows RWS 500 over the latter half of the 21st century in CMIP5 HIGH-r models.

Reviewer #2:

I am generally supportive of publication of this work and wish to encourage the authors' efforts in this important area. However, I do have substantial reservations about the level of certainty with which the conclusions are phrased, and feel that the study would benefit from improvements in the way it is rooted in prior literature on this topic.

This study is a significant contribution in terms of following up on the question raised in reference 5 regarding how changes in the Eastern Pacific jetstream and storm track coming onto California affects California precipitation, especially in light of how the region since the transition between the tendency towards increased precipitation to the north and decreased precipitation to the south. This hypothesis was motivated by the principal mechanism by which El Niño affects California rainfall, so that study used El Niño interannual variability to check the precipitation to Eastern-Pacific jet relations. And because long-term El Niño-like change is one of the most obvious hypotheses for the jet impacts on precipitation change in California, that study also explicitly checked for a correlation to El Niño indices, such as Niño 3.4 SST, across the portion of the CMIP5 ensemble that was available. Niño 3.4 SST per se was not found to be a strong correlation but large-scale changes in the jet occur under global warming independent of equatorial cold tongue SST were found to yield a pathway similar to the El Niño effect of stirring additional storms onto the coast. That study left the exact source of the jetstream changes undetermined, so mechanistic studies such as the ones presented in the current Allen and Luptowitz manuscript can be highly valuable in determining the physical source of these changes. But it would be beneficial to more clearly indicate what has already been examined. [In passing, it's worth noting that there is substantial parallels in parts of the abstract of the present manuscript and reference 5 that might be worth referencing within the abstract itself since this is permitted in Nature journals].

We have placed more emphasis on prior work in the Introduction, including adding some of the above text to the Introduction. Nature Communications does not allow citations in the Abstract. There is also a 150 word limit for the Abstract, so adding to it is not possible (we actually had to trim it down).

A subsequent study, Langenbrunner et al. (2015, J. Climate) [not currently cited in the manuscript] looked at the CMIP5 model ensemble uncertainty in this region, and also explicitly sought an explanation in tropical Pacific sea surface temperature. It found that the leading mode affecting the inter-model range of projections for California rainfall change under global warming was related to a more complex large-scale pattern of SST changes with little expression in the equatorial cold tongue region. A second mode of North Pacific intermodel precipitation change differences was related to SST patterns that included a substantial El Niño like component, but this had relatively modest expression in California rainfall. Thus the model response that affects both the jet and California rainfall across the CMIP5 ensemble appears to have other contributions than El Niño like changes. It would be appropriate for the present study to frame itself with respect to this prior work; it would then be clear the hypothesis has been previously examined at the basic level of relations across the CMIP5 ensemble and would highlight the question of what aspect of the current CMIP5 analysis suggests an El Niño like contribution can be significant for some part of the model ensemble.

We have added much of this information to the Introduction, and now cite Langenbrunner et al. 2015.

The answer to this question in the present manuscript is conditioning on high correlation between California rainfall and El Niño SST indices in interannual variability. The authors show that an average over a subset of models with high interannual CA-Precip–equatorial-Pacific-SST correlation (“high-r”) tends to exhibit larger California precipitation changes relative to an average over a low-correlation subset. This is a nice analysis and a very reasonable way to parse out the results in a way that distinguishes certain behaviors.

However, high-r alone does not guarantee a more realistic response. Overly high correlation would also be unrealistic and high correlation can go with unrealistic amplitude response. Some models exhibit larger-than-observed regressions of precipitation on Niño indices in this region. This was true of the previous version of the NCAR model (e.g., Langenbrunner et al. 2013), and may well be inherited by the version used here. Furthermore, many models have a storm track that hits the US West Coast too far south. For such models there is considerable concern that increases in California are overestimated, although the increase at the end of the storm track may be qualitatively correct but simply misplaced in latitude along the coast. Such models can easily exhibit excessive rainfall sensitivity in California. If one took a subset of CMIP5 models conditioned on an interannual regression coefficient within the error bars of the observational coefficient, and a condition for a correct climatological precipitation latitude dependence in California within the observational error would this subensemble yield the same results? The high correlation alone is not sufficient to fully support statements such as “models that better simulate the observed El Niño-CA precipitation teleconnection” (lines 13-14), but inclusion of a measure of regression coefficient and climatological precipitation would substantially help.

We have added much of this discussion to the manuscript and Supplement, including a new analysis that looks at the DJF CA precipitation versus Niño 3.4 SST regression coefficient, and DJF CA precipitation climatologies. The following text has been added to the manuscript:

We also note that high correlation between CA precipitation and Niño 3.4 SSTs alone does not guarantee a more realistic response. Overly high correlation would also be unrealistic, and high correlation can go with unrealistic amplitude response. Some models exhibit larger than observed regressions of CA precipitation onto Niño 3.4 SSTs (Langenbrunner and Neelin 2013). Furthermore, some models have a storm track that hits the US West Coast too far south. Supplementary Table 2 shows late-20th century DJF correlations, regression coefficients, and climatological CA precipitation by region for CMIP5 models, along with the observed values and their 1-sigma uncertainty. The CA precipitation sensitivity to Niño 3.4 SSTs, and regional climatologies, for many models falls outside the observed range (Supplement). We define an alternative “HIGH-r” model subset that satisfies the following criteria: 1. late-20th and 21st century correlations between DJF CA precipitation and Niño 3.4 SSTs are significant at the 90% confidence level; 2. late-20th century DJF CA precipitation versus Niño 3.4 SST regression coefficient falls within 1-sigma of the observed range; and 3. late-20th century DJF CA precipitation climatologies fall within 1-sigma of the observed range. The ensemble mean 21st century DJF (ANN) CA precipitation trend in these HIGH-r models is 0.91 (0.24) mm day⁻¹ century⁻¹, both significant at the 99% confidence level. 83% (83%) of the realizations yield an increase in DJF (ANN) CA precipitation. These values are very similar to those based on our

original correlation-based threshold. Moreover, the tropical and extratropical dynamical responses in these three models is similar to that discussed in the manuscript (not shown). Thus, we continue to find significant and robust increases in 21st century CA precipitation, particularly in those models that better simulate CA precipitation statistics

In the SST experiments, similar concerns arise. Both the GFDL model and the NCAR large ensemble exhibit versions of this issue in the position of the southern boundary of the storm track where it arrives at the US West Coast, tending to extend too far south. This does not invalidate qualitative results using those models here, but it does indicate substantial caveats that should be acknowledged.

We now acknowledge that these two models overestimate the sensitivity of CA precipitation to Niño 3.4 SSTs (e.g., Supplementary Table 2). GFDL in particular also tends to overestimate CA precipitation climatologies. This has been added to the revision.

The experiments in Figure 7 are not convincing because the SST gradients are so much larger than the range of anything that would ever occur: the experiment with no warming in Niño 3.4 SST is similar to having a 3 or 4° La Niña relative to an overall warm climate. And the question to be addressed is not whether A) an enormous El Niño or La Niña can have an impact on California, but whether B) the relatively modest equatorial temperature variations relative to the overall warmer climate adds anything significant to the changes to the jetstream that would occur without these, associated with larger scale warming. Figure 7 shows the former — which is unsurprising and serves as a red herring in terms of the flow of the argument — but does not add to the evidence for the latter. Supplementary figures 11-12 suffer from this same issue. Figure S13 in the supplement shows an experiment that does appear to better address this issue in that it compares a patterned SST case to a uniform SST warming case. This doesn't distinguish between large-scale tropical-extratropical gradients and local Pacific enhancements, but it does at least distinguish from a uniform warming.

We have replaced these experiments with new simulations that isolate the entire tropical Pacific (as well as the entire tropics, etc.), as opposed to the Niño3.4 SST region. We find similar results. We believe these new simulations better support our argument that warming of tropical Pacific SSTs is important for the increase in CA precipitation.

Supp. Figure 11 and 12 have been removed/replaced.

We have also moved the discussion of AMIP4K and AMIP Future simulation from the Supplement to the main text (Nature Communications allows up to 5000 words and 10 display items). We also move Supplementary Figure 13, which shows the precipitation response in AMIP models, to the main text (Figure 7 now).

The description of these SST experiments could be more clearly written — it takes a while to parse out what is specifically being compared in each case. But I would recommend swapping figure S13 from the supplement into the main text, and moving figure 7 into the supplement. The results that are alluded to in line 218-219 "idealized CAM5 simulations forced with uniform tropical Pacific warming also reproduce these results (not shown)" are very intriguing. They seem to indicate that amplified warming in the equatorial Pacific is not essential, consistent with

examination of the equatorial Pacific effect in the earlier studies. It would be worth including these experiments. That would suggest that a broad scale tropical warming could affect the jet stream and that would still be consistent with models with higher El Niño response in California reacting more strongly to this jetstream change.

We have edited/cleaned-up the section that describes our idealized experiments (in Methods).

As mentioned above, we have added these uniform tropical Pacific SST warming experiments to the manuscript. They are now shown in Figure 8. Corresponding GFDL results are included in the Supplement. We also conduct additional tropical warming experiments (as described in the new Methods section) that support these results.

Turning to the conclusions, in both the abstract line 15 “Our results suggest that California will likely become wetter in a warmer world.” and at the end of the discussion line 232 “[these results] imply that California will likely become wetter in response to continued greenhouse gas emissions.” These statements seem overly confident in light of the model uncertainties and of the prior analysis discussed above. California precipitation projections are a serious matter, with real consequences for infrastructure decisions and for public confidence in climate science. Both the “likely” and “wetter” are aspects of this phrasing that are open to misinterpretation. While the evidence here is useful in moving the understanding of these projections forward, it elucidates a factor rather than leading to an overall conclusion of likely increases. Furthermore, “wetter” refers only to the precipitation, whereas there is a substantial contribution to the water budget as experienced on the ground due to evapotranspiration increases (e.g. reference 24). This is not addressed here, and yet would be important factor before one could make a sweeping statement about wetter climate.

We have toned down our claims that CA will likely become wetter. We do note, however, robust increases in CA precipitation in the three models that satisfy all of the criteria mentioned above (based on correlations, sensitivities, and climatologies). We now state that CA may receive more precipitation in the future, and acknowledge model uncertainties.

I'd like to make suggestions on how to revise these conclusions, but first let me turn to some strengths of this paper that deserve highlighting. In the dynamical analysis lines 114-182, the results are very coherent. The findings regarding meridional wind versus zonal wind help to distinguish between two dynamical hypotheses. The authors do a nice job of bringing together storm track activity measures from reference 9 [although they could be a little clearer in attributing these measures] with wind measures and moisture convergence measures to provide a thorough picture of dynamical changes associated with the high-r models.

Revising the conclusions to adhere more closely to what has been specifically shown: there is a dynamically coherent response, consistent with previous analysis, involving a southeastward shift of the upper level winds and an increase in storm track activity in the east Pacific, and an increase in CA moisture convergence and that models with higher correlation between California precipitation and El Niño interannual variability yield larger and more consistent increases in California precipitation under global warming. And that while local enhancement of cold tongue SST does not appear to be essential to the changes in the jetstream and associated, or precipitation changes, there are indications that they can enhance it.

We have revised the conclusions (now a discussion section) to be more in line with what is discussed above.

REVIEWERS' COMMENTS:

Reviewer #1 (Remarks to the Author):

I appreciate the authors work to address the comments in my third review. The revised paper includes a more cautious tone regarding the results ("will" versus "may") and a more careful discussion of previous work. It represents a substantial improvement.

Reviewer #2 (Remarks to the Author):

These revisions do address essentially all the suggestions (and concerns) I had regarding this manuscript and seem to me to make it clearer, more consistent with prior literature and more carefully bounded in framing the results. I recommend publication.

The only remaining optional suggestion I would make is that in recapping the SST experiments, a brief phrase be included to remind readers less familiar with this type of experiment that the hypothetical situation of the tropical Pacific not warming as the rest of the global SST does is simply a technique to partition regional effects of the overall warming in different locations.

(signed)

David Neelin

Response to Reviewers

Reviewer #1 (Remarks to the Author):

I appreciate the authors work to address the comments in my third review. The revised paper includes a more cautious tone regarding the results (“will” versus “may”) and a more careful discussion of previous work. It represents a substantial improvement.

We again thank Reviewer #1 for evaluating our paper.

Reviewer #2 (Remarks to the Author):

These revisions do address essentially all the suggestions (and concerns) I had regarding this manuscript and seem to me to make it clearer, more consistent with prior literature and more carefully bounded in framing the results. I recommend publication.

The only remaining optional suggestion I would make is that in recapping the SST experiments, a brief phrase be included to remind readers less familiar with this type of experiment that the hypothetical situation of the tropical Pacific not warming as the rest of the global SST does is simply a technique to partition regional effects of the overall warming in different locations.

(signed)

David Neelin

We thank David Neelin for his helpful comments. We have added a sentence to the revised manuscript that explains the tropical Pacific SST warming experiments are a technique used to isolate the role of certain regions.